# SNX19 restricts endolysosome motility through contacts with the endoplasmic reticulum

Amra Saric[1], Spencer A. Freeman[2,3], Chad D. Williamson[1], Michal Jarnik [1], Carlos M. Guardia [1], Michael S. Fernandopulle[4], David C. Gershlick [1,5] & Juan S. Bonifacino [1✉]

The ability of endolysosomal organelles to move within the cytoplasm is essential for the performance of their functions. Long-range movement involves coupling of the endolysosomes to motor proteins that carry them along microtubule tracks. This movement is influenced by interactions with other organelles, but the mechanisms involved are incompletely understood. Herein we show that the sorting nexin SNX19 tethers endolysosomes to the endoplasmic reticulum (ER), decreasing their motility and contributing to their concentration in the perinuclear area of the cell. Tethering depends on two N-terminal transmembrane domains that anchor SNX19 to the ER, and a PX domain that binds to phosphatidylinositol 3-phosphate on the endolysosomal membrane. Two other domains named PXA and PXC negatively regulate the interaction of SNX19 with endolysosomes. These studies thus identify a mechanism for controlling the motility and positioning of endolysosomes that involves tethering to the ER by a sorting nexin.

[1] Neurosciences and Cellular and Structural Biology Division, Eunice Kennedy Shriver National Institute of Child Health and Human Development, National Institutes of Health, Bethesda, MD, USA. [2] Program in Cell Biology, Peter Gilgan Centre for Research and Learning, Hospital for Sick Children, Toronto, ON, Canada. [3] Department of Biochemistry, University of Toronto, Toronto, ON, Canada. [4] Neurogenetics Branch, National Institute of Neurological Disorders and Stroke, National Institutes of Health, Bethesda, MD, USA. [5] Present address: Cambridge Institute for Medical Research, University of Cambridge, Cambridge, UK. ✉email: juan.bonifacino@nih.gov

Endosomes and lysosomes (henceforth jointly referred to as "endolysosomes" or "ELs") are membrane-bound organelles that primarily mediate uptake, degradation and recycling of both exogenous materials delivered by endocytosis and endogenous materials delivered by biosynthetic transport or autophagy[1,2]. In addition, ELs participate in many other cellular processes, including the regulation of cell metabolism, proliferation, adhesion and migration, immunity, exocytosis, and plasma membrane repair[3,4]. These diverse functions of ELs are highly dependent on their ability to move throughout the cytoplasm[5]. EL motility is mediated by interactions with motor proteins such as kinesins and dynein-dynactin, which drive movement along microtubules in the plus-end (i.e., anterograde) and minus-end (i.e., retrograde) directions, respectively[5,6]. This movement is influenced by interactions of ELs with other organelles, mainly the endoplasmic reticulum (ER)[5,6]. Interactions with ELs occur at specific domains of the organelles known as membrane contact sites (MCS)[7]. One type of ER–EL interaction that influences EL motility involves the ER-anchored ubiquitin ligase RNF26, which recruits and ubiquitinates the cytosolic sequestosome 1 protein SQSTM1, enabling binding to various EL adaptors having ubiquitin-binding domains such as EPS15, TAX1BP1, and TOLLIP[8]. Another type is represented by the ER-anchored protein protrudin (also known as ZFYVE27), which binds ELs through coincident detection of the small GTPase RAB7 and the phosphoinositide phosphatidylinositol 3-phosphate [PI(3)P] on the EL surface[9]. Finally, a third type involves the ER-anchored protein VAP (VAP-A and VAP-B paralogs), which interacts with the endolysosomal cholesterol-sensing protein ORP1L[10]. ER–EL tethering mediated by each of these three systems is reversed by the action of the deubiquitinating enzyme USP15[8], the adaptor protein FYCO1[9], and high cholesterol[10], respectively, freeing the ELs for interaction with microtubule motors. The balance between interactions with microtubule motors and ER-tethering factors determines the overall movement and distribution of ELs within the cytoplasm.

Studies in yeast identified Mdm1 as another protein that bridges the nuclear envelope (a domain of the ER) with the vacuole (the functional equivalent of the metazoan lysosome) at MCS known as nuclear–vacuole junctions (NVJ)[11]. Further studies showed that yeast Mdm1 also associates with lipid droplets (LDs), clustering LDs next to NVJ and thus contributing to LD expansion under conditions of increased fatty acid availability[12]. *Drosophila melanogaster* has a single Mdm1 ortholog named snazarus or SNZ, which also regulates LD homeostasis, although in this case at MCS between the ER and the plasma membrane[13]. Mammals have 4 Mdm1/SNZ homologs named SNX13, SNX14, SNX19, and SNX25[11,14] (Fig. 1a). All of these proteins belong to a subset of the larger family of sorting nexin (SNX) proteins, which are defined by the presence of a phox-homology (PX) domain[14] (Fig. 1a). In addition to a PX domain, this subset comprises two predicted N-terminal transmembrane (TM) domains, a PX-associated domain (PXA), and a C-Nexin domain (PXC) (Fig. 1a)[11,14]. The best characterized of the mammalian Mdm1/SNZ orthologs is SNX14[15,16]. Like the yeast and Drosophila orthologs, SNX14 localizes to ER-LD contact sites and promotes fatty acid-mediated LD growth[16]. Whereas the role of Mdm1, SNZ, and SNX14 in lipid homeostasis is now well established, to date there is no evidence that SNX14 or the other mammalian orthologs mediate tethering of the ER to lysosomes or other endolysosomal organelles, much less that they regulate EL positioning.

Herein we report that SNX19, but not SNX14, promotes ER–EL contacts and that these contacts restrict EL motility. Contacts are dependent on the two N-terminal TM domains that anchor SNX19 to the ER membrane and the PX domain that binds PI(3)P on the EL membrane. Furthermore, we show that the PXA and PXC domains inhibit the ability of SNX19 to mediate ER–EL tethering, indicating that they participate in an auto-regulatory mechanism. SNX19-mediated tethering to the ER decreases EL motility and contributes to the maintenance of a perinuclear population of ELs. Finally, we show that SNX19 can be additionally recruited to ER-LD contact sites in the presence of excess fatty acids. SNX19 thus has the properties of a protein capable of mediating both ER–EL and ER-LD tethering.

## Results

**SNX19 localizes to endoplasmic reticulum–endolysosome contact sites.** SNX13, SNX14, SNX19, and SNX25 constitute a subfamily of mammalian sorting nexins that contain two predicted N-terminal TM domains, followed by PXA, PX, and PXC domains (Fig. 1a)[11,14]. In addition, SNX13, SNX14, and SNX25 have a regulator of G-protein signaling (RGS) domain that is absent in SNX19 (Fig. 1a)[11,14]. The best-characterized member of this family is SNX14, which localizes to ER-LD contact sites and regulates LD growth upon fatty acid addition[15,16]. In contrast, the localization and function of SNX19 are poorly understood. Besides the lack of an RGS domain, SNX19 differs from SNX14 in that the SNX19 PX domain binds PI(3)P whereas the SNX14 PX domain does not bind any phosphoinositide[17,18]. The fact that PI(3)P is enriched in ELs[19] prompted us to investigate if SNX19 could mediate ER–EL contacts.

Although SNX19 is ubiquitously expressed (http://www.proteinatlas.org), we were unable to determine the intracellular localization of the endogenous protein by immunofluorescence microscopy of various human cell lines using commercially available antibodies, probably due to the low abundance of this protein (0.59 ppm; for comparison, LAMP1 abundance is 115 ppm; https://pax-db.org/). To overcome this limitation, we transfected a plasmid encoding SNX19 tagged with a C-terminal green fluorescent protein (GFP) (SNX19-GFP) into human osteosarcoma U-2 OS cells, which are large, flat, and thus highly amenable to light microscopy analyses. Confocal microscopy of fixed cells showed localization of SNX19-GFP to a network that was identified as the ER by co-staining for the ER marker calnexin (Fig. 1b). With the exception of very high expressors, transfected and untransfected cells did not show appreciable differences in the overall structure and distribution of the ER network. Deletion of the two predicted TM domains resulted in localization of the truncated protein (SNX19$^{\Delta TM}$-GFP) to the cytosol (Fig. 1c), demonstrating that SNX19 is anchored to the ER via the TM domains.

A similar pattern of localization to the ER was observed in live U-2 OS cells expressing Halo-tagged SNX19 (Fig. 1d, left panels). However, closer inspection of these live cells revealed the presence of higher intensity SNX19-Halo puncta along with the ER (Fig. 1d, left panels, arrowheads). To determine whether these puncta represented contacts with other organelles, we co-expressed SNX19-Halo with various fluorescent organelle markers and assessed their co-localization by confocal microscopy. We observed that SNX19-Halo puncta were closely apposed to a subset of organelles containing the EL marker LAMP1-RFP[20] (Fig. 1d). Additional experiments showed apposition of transgenic SNX19-GFP puncta with endogenous LAMP1 and early endosomal marker EEA1 (Supplementary Fig. 1a). In contrast, we did not observe a significant association of SNX19-Halo with the LD marker GFP-ADRP[21] or the mitochondrial marker DsRed-Mito[22] under normal conditions of culture (Fig. 1d). Time-lapse imaging of live cells co-expressing SNX19-GFP and LAMP1-RFP showed that both signals remained together over time, irrespective of whether they appeared static (most cases) or motile (fewer

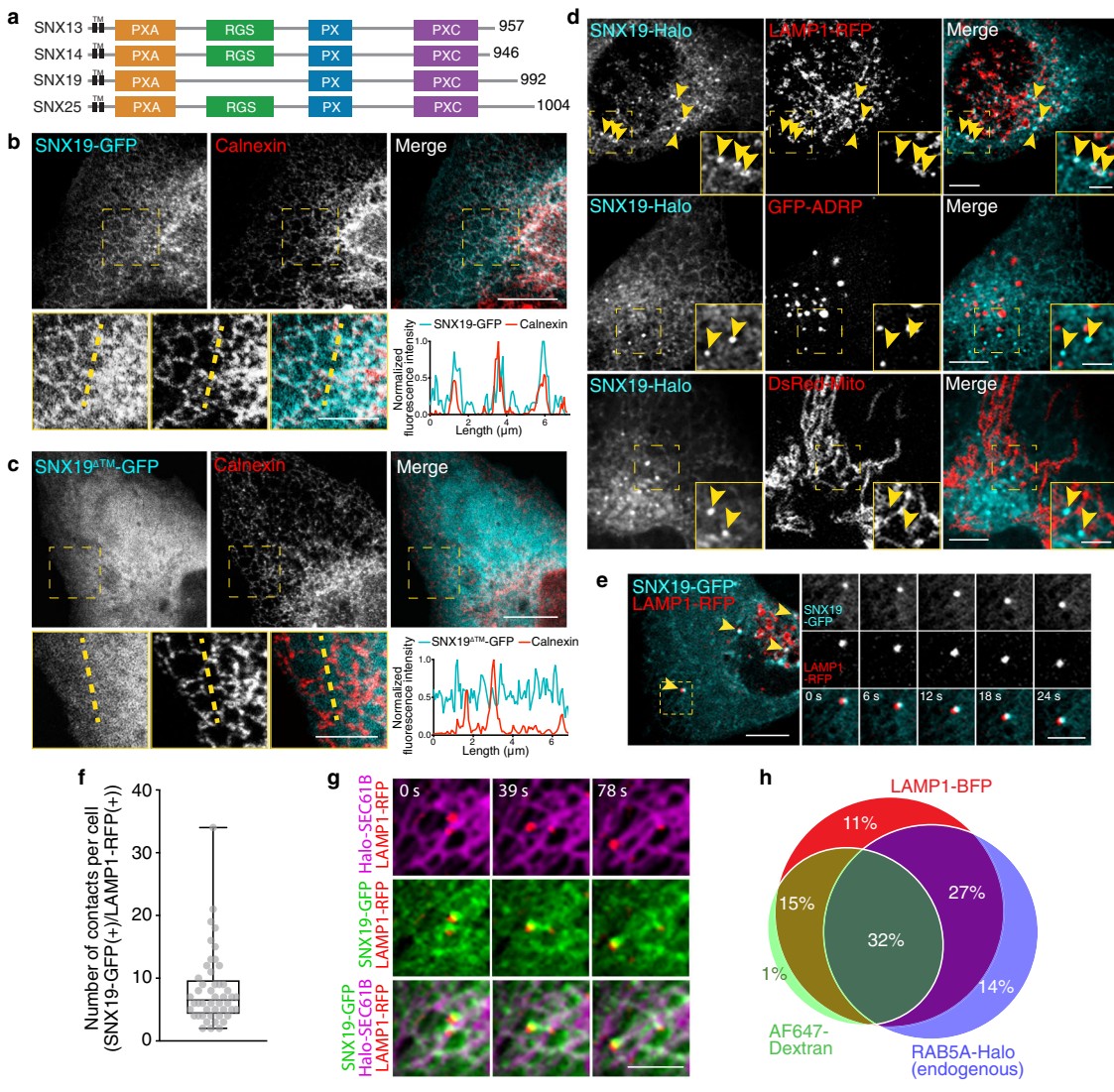

**Fig. 1 SNX19 is an ER-resident protein that makes contacts with endolysosomes. a** Scheme of mammalian SNX19 and its paralogs. TM transmembrane domains, PXA PX-associated domain, RGS regulator of G-protein signaling domain, PX phagocyte oxidase (phox) homology domain, PXC C-Nexin domain. Amino-acid numbers of each protein are indicated on the right. Scheme not to scale. **b, c** U-2 OS cells were transfected with plasmids encoding SNX19-GFP (**b**) or SNX19ΔTM-GFP (lacking the TM domains) (**c**), fixed, immunostained with antibody to calnexin, and imaged by confocal microscopy. The bottom rows show zoomed in images of the indicated areas at higher contrast. Fluorescence intensity along the yellow dashed line was measured, normalized by setting the highest value for each channel to 1 and scaling all other values, and represented as a line scan (bottom right in each panel). **d** U-2 OS cells were co-transfected with plasmids encoding SNX19-Halo (subsequently labeled with Janelia Fluor 646 dye) and either LAMP1-RFP (ELs), GFP-ADRP (LDs), or DsRed-Mito (mitochondria), and imaged live by confocal microscopy. Yellow arrowheads indicate representative SNX19-Halo puncta. **e** U-2 OS cells were transfected with plasmids encoding SNX19-GFP and LAMP1-RFP and analyzed by time-lapse, live-cell imaging (Supplementary Movie 1). Arrowheads point to representative SNX19-GFP–LAMP1-RFP contacts in one cell. Small images are frames from a video of the dashed box in the large image, showing individual and merged channels over time. **f** Number of puncta positive for SNX19-GFP and LAMP1-RFP per cell was quantified from 48 cells across 5 live-cell imaging experiments conducted as in **e**. The data were represented as a dot plot of individual values overlaid with a box and whiskers plot with the box extending from 25th to 75th percentiles, the horizontal line representing the median, and the whiskers representing the minimum and maximum values. **g** U-2 OS cells were co-transfected with plasmids encoding Halo-SEC61B (subsequently labeled with Janelia Fluor 646 dye), SNX19-GFP, and LAMP1-RFP, and imaged live by confocal microscopy. The images show a SNX19-positive EL and the ER over time. **h** HeLa RAB5A-Halo-KI cells were co-transfected with plasmids encoding SNX19-GFP and LAMP1-BFP, and incubated for 16 h with Alexa Fluor 647-Dextran followed by a 2-h chase, endogenously tagged RAB5A-Halo (early endosome marker) was labeled with Janelia Fluor 549 dye, and cells were imaged live by confocal microscopy. The percentage of SNX19-GFP puncta that contained one or more of the markers was calculated and represented as a Venn diagram. A total of 135 contacts across 2 experiments, 15–20 cells per experiment, were quantified. Source data are provided as a Source Data file. Scale bars: 10 μm in **b**–**e**, 5 μm insets and in **g**.

cases) (Fig. 1e and Supplementary Movie 1), indicating that these puncta represent *bona fide* ER–EL associations. Quantification of puncta containing both SNX19-GFP and LAMP1-RFP and that co-localized for at least 10 s in live cells (a rather strict requirement) revealed an average of 8 contacts per cell, with 25% of cells exhibiting between 10 and 35 contacts per cell, at any given time (Fig. 1f). Further live-cell imaging showed that SNX19-GFP foci associated with LAMP1-RFP were continuous with a fainter SNX19-GFP signal distributed throughout the ER, identified by labeling with Halo-tagged SEC61B (Fig. 1g).

Overexpressed LAMP1 has been shown to localize not only to lysosomes but also to early and late endosomes[23]. Thus, to better

define the identity of the EL organelles that make contacts with SNX19, we performed four-color fluorescence microscopy of live HeLa cells expressing the early endosomal marker RAB5A endogenously labeled with the Halo tag [i.e., RAB5A-Halo knock-in (KI) cells] (Supplementary Fig. 1b), which were additionally transfected with plasmids encoding SNX19-GFP and LAMP1-BFP, and incubated live for 16 h with the fluid-phase endocytic marker Alexa Fluor 647 (AF647)-Dextran followed by a 2 h chase in regular medium (Supplementary Fig. 1c). Quantification of the co-localization with these markers showed that the majority of SNX19-GFP contacts occurred with organelles having all three markers (32%), or combinations of LAMP1-BFP plus RAB5A-Halo (27%), or LAMP1-BFP plus AF647-Dextran (15%), which we took to represent late endosomes, early endosomes and lysosomes, respectively (Fig. 1h and Supplementary Fig. 1d).

From these experiments, we concluded that: (1) SNX19 mediates contacts of the ER mainly with early/late endosomes, and, to a lesser extent, terminal lysosomes, and (2) LAMP1 is a valid pan-EL marker for identifying contacts with SNX19.

## SNX19 contacts endolysosomes via interaction of its PX domain with PI(3)P.

To investigate the mechanism by which SNX19 contacts ELs, we considered the possibility that a specific SNX19 domain could recognize EL membrane lipids. While little is known about the PXA and PXC domains, the PX domain of SNX19 has been shown to bind PI(3)P[17,18]. Indeed, assays using PIP Strips confirmed specific binding of the recombinant GST-tagged PX domain of SNX19 (GST-PX; Fig. 2a) to PI(3)P, and to a lesser extent, phosphatidylserine (PS) (Fig. 2b). Moreover, mutation of a PX domain residue predicted to be involved in PI(3)P binding[17] (R582 to Q) abrogated binding to PI(3)P on the PIP Strips (Fig. 2a, b; GST-PX$^{R582Q}$).

To examine a possible role of PI(3)P recognition in the association of SNX19 with ELs, we examined the distribution of full-length SNX19-GFP with mutation of R582 to Q. We observed that this mutant construct still localized to the ER network but showed reduced association with LAMP1-RFP-positive puncta in U-2 OS cells (Fig. 2c, d). In addition, we tested the effects of drugs that decrease or increase intracellular PI(3)P levels. To deplete PI(3)P, we treated cells with SAR405, an inhibitor of VPS34, the lipid kinase that converts PI to PI(3)P on endosomes (Fig. 2e)[24]. The effectiveness of this inhibitor in our cell system was demonstrated by its ability to dissociate the PI(3)P probe 2xFYVE-GFP from ELs (Supplementary Fig. 2). We found that SAR405 treatment decreased the number of SNX19-GFP puncta that co-localized with LAMP1-RFP (Fig. 2f, g). Conversely, to increase PI(3)P levels we used apilimod, an inhibitor of PIKFYVE, the lipid kinase that converts PI(3)P to PI(3,5)P$_2$[25] (Fig. 2e). Treatment with apilimod caused EL swelling (Fig. 2f), as previously reported[25]. Importantly, this treatment resulted in a dramatic increase in the number of SNX19-GFP puncta that were associated with the swollen LAMP1-RFP-positive ELs (Fig. 2f, g).

Taken together, these results indicated that SNX19-EL contacts are mediated by binding of the SNX19 PX domain to PI(3)P on the EL membrane.

## Deletion of the SNX19 PXA or PXC domains increases association with endolysosomes.

Since little is known about the PXA and PXC domains, we investigated their role in the interaction of SNX19 with ELs. To this end, we generated SNX19 deletion mutants lacking either the PXA domain (SNX19$^{\Delta PXA}$-GFP) or a C-terminal segment containing the PXC domain (SNX19$^{1-659}$-GFP) (Fig. 3a). Confocal microscopy showed that both SNX19$^{\Delta PXA}$-GFP and SNX19$^{1-659}$-GFP had a significantly larger number of contacts with LAMP1-RFP-positive ELs relative to full-length SNX19-GFP (Fig. 3b,

c). Foci containing full-length or mutant SNX19-GFP together with LAMP1-RFP were always associated with the ER network, as visualized by co-labeling with a BFP-tagged KDEL ER-retrieval signal (Fig. 3d) or Halo-tagged VAP-A tethering protein (Supplementary Fig. 3a–c). The latter two ER proteins, however, were not visibly concentrated at SNX19-GFP–LAMP1-RFP contact sites (Fig. 3d and Supplementary Fig. 3a–c). These observations indicated that the PXA and PXC domains inhibit the interaction of SNX19 with ELs.

The inhibitory effect of these domains likely explains why a SNX19-GFP mutant lacking the two transmembrane domains localizes to the cytosol (Fig. 1c), despite having a PX domain intrinsically capable of binding PI(3)P on the EL membrane (Fig. 2). Indeed, deletion of the PXA domain in addition to the transmembrane domains resulted in a SNX19-GFP construct that localized to a subset of LAMP1-RFP-positive vesicles (Supplementary Fig. 3d). We termed SNX19 mutants lacking the PXA or PXC domains "hypertethers" because of their enhanced ability to tether ELs to the ER.

The identification of the SNX19 hypertethers greatly facilitated the ultrastructural visualization of SNX19-EL contacts by correlative light and electron microscopy (CLEM). In these experiments, cells expressing SNX19$^{\Delta PXA}$-GFP or SNX19$^{1-659}$-GFP together with LAMP1-RFP and Mito-BFP (mitochondrial marker) were allowed to endocytose BSA-gold for 24 h, fixed and imaged by high-resolution structured illumination microscopy (SIM) (Fig. 4a, c), after which selected contacts were analyzed by transmission EM (Fig. 4b, d and Supplementary Fig. 4a, b). Labeling mitochondria with Mito-BFP served two important purposes: (1) it facilitated the manual alignment of fluorescence and EM images, as these organelles are large and easily identifiable landmarks, and (2) they were used as an internal specificity control for SNX19 contacts, which occurred with LAMP1-RFP but not Mito-BFP labeled organelles. Using this approach, we could observe ER cisternae (colored green) that were tightly apposed to organelles with the morphology of ELs at sites of enrichment of both hypertethers (yellow arrowheads) (Fig. 4b, d and Supplementary Fig. 4a, b). Three-D reconstruction from serial sections revealed the extensive nature of ER–EL contacts at those sites (Supplementary Fig. 4c). CLEM also showed that the LAMP1-RFP-positive EL compartment enriched in hypertethers most often had the morphology of a multi-vesicular body (MVB), as intraluminal vesicles could be readily observed in their interior (Fig. 4b, d and Supplementary Fig. 4a). The CLEM image in Fig. 4b additionally shows that a neighboring LAMP1-RFP-positive structure lacking SNX19$^{\Delta PXA}$-GFP and containing internalized BSA-gold (black arrowhead) has the morphology of a lysosome. These observations supported our earlier finding that SNX19 contacts mostly endosomes, and to a lesser extent terminal lysosomes (Fig. 1h and Supplementary Fig. 1d). LDs, identified as small, electron-dense (i.e., osmiophilic) bodies were often found in the proximity of SNX19-decorated ELs, but the LDs themselves did not have associated SNX19 under the conditions of these experiments (Fig. 4b and Supplementary Fig. 4a, white arrowheads).

## Loss of SNX19 decreases both contacts of endolysosomes with the ER and perinuclear distribution of endolysosomes.

Next, we examined the contribution of SNX19 to the overall extent of ER–EL contacts within the cell. To this end, we knocked out the gene encoding SNX19 in U-2 OS cells using CRISPR–Cas9 (Supplementary Fig. 5a–c) and analyzed the effect of the KO on ER–EL contacts using two methods. In the first method, we allowed cells to endocytose BSA-gold for 24 h to label ELs, and then examined serial sections of plastic-embedded cells by transmission EM for the presence of ER–EL contacts defined by

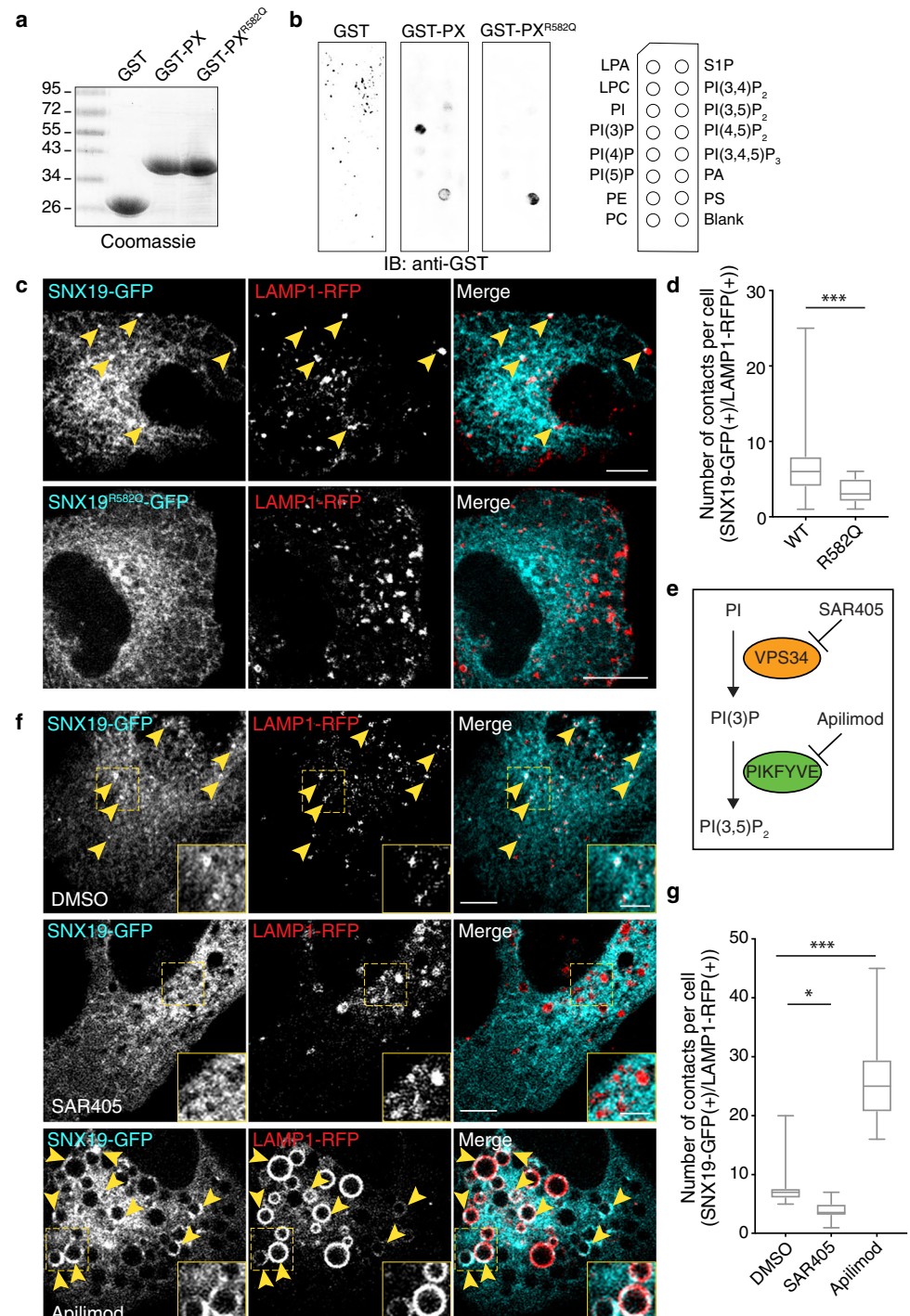

**Fig. 2 SNX19 contacts endolysosomes via interaction of its PX domain with PI(3)P. a** Purified GST, GST-PX domain from SNX19 and GST-PX$^{R582Q}$ containing a point mutation in the PX domain of SNX19, were subjected to SDS-PAGE, and the gel was stained with Coomassie blue. The positions of molecular mass markers (in kDa) are indicated on the left. Experiment was repeated two independent times. **b** PIP Strips were incubated with purified GST, GST-PX, or GST-PX$^{R582Q}$, and immunoblotted for GST. A scheme of spotted lipids on the strips is shown at right. **c** U-2 OS cells were co-transfected with expression plasmids encoding either SNX19-GFP or SNX19$^{R582Q}$-GFP together with LAMP1-RFP, and imaged live by confocal microscopy. Arrowheads in the upper row point to representative SNX19-EL contacts. **d** Quantification of SNX19-GFP–LAMP1-RFP contacts from **c**. Box and whisker plots (drawn as indicated for Fig. 1f) are from 25 to 40 cells per condition, across $n = 3$ experiments. Mann–Whitney test was applied to determine significance, *** two-tailed $p$-value < 0.001. **e** Scheme showing lipid kinase inhibitors and their targets. **f** U-2 OS cells were co-transfected with plasmids encoding SNX19-GFP and LAMP1-RFP, treated with either DMSO control (top row), 1 μM SAR405 (middle row), or 200 nM apilimod (bottom row) for 2 h, and imaged live by confocal microscopy. Arrowheads point to representative SNX19-GFP–LAMP1-RFP contacts. **g** Quantification of contacts observed in **f**. Box and whisker plots represent data from three independent experiments, where 10–20 cells were quantified per condition, per experiment. Significance was analyzed by Kruskal–Wallis one-way ANOVA; *$p = 0.0175$, ***$p < 0.001$. For plots in **d** and **g**, the box extends from 25th to 75th percentiles, the horizontal line represents the median, and the whiskers represent the minimum and maximum values. Source data are provided as a Source Data file. Scale bars: 10 μm in **c** and **f** 5 μm insets.

apposition of the two organelle membranes at ≤30 nm[26] (Fig. 5a). Quantification of the EL perimeter length engaged in contacts with ER elements relative to the total EL perimeter length revealed a statistically significant ~2-fold decrease in the extent of ER–EL contacts in SNX19-KO relative to WT cells (Fig. 5b).

The second method was a proximity ligation assay (PLA)[27] using primary antibodies to endogenous LAMP1 and calnexin, followed by PLA probes. Immunofluorescence microscopy showed fewer fluorescent puncta in SNX19-KO relative to WT cells (Fig. 5c, d), indicative of a reduction in the overall number of ER–EL contacts.

Having established that SNX19 contributes to overall ER–EL contacts, we next examined whether these contacts influence the cytoplasmic distribution of ELs. Immunostaining for various organelle markers revealed that the LAMP1-positive EL population in SNX19-KO cells was more dispersed than in wild-type (WT) cells (Fig. 5e). Quantification of this dispersal using a semi-automated shell analysis of 60–90 cells per condition showed that the fraction of peripheral ELs rose from a median of 0.15 in WT cells to 0.42 in SNX19-KO cells (Fig. 5f). Re-expression of SNX19-GFP in SNX19-KO cells partially reversed the EL dispersal phenotype (Fig. 5e, f), confirming that this phenotype was due to the loss of SNX19. In contrast to the distribution of ELs, the distribution of the ER, the Golgi complex, and mitochondria immunolabeled with antibodies to calnexin, GM130, and Tomm20, respectively, was not affected by SNX19 KO (Supplementary Fig. 5d–f). These experiments thus demonstrated that SNX19-mediated ER–EL contacts contribute to the maintenance of the perinuclear population of ELs.

**SNX19 restricts endolysosome motility**. The EL dispersal phenotype of SNX19-KO cells suggested that SNX19-mediated ER–EL contacts constrain ELs to the perinuclear region. To determine whether this phenomenon was due to an effect on EL motility, we performed live imaging of ELs in U-2 OS cells at ~10 fps, followed by analysis using automated particle detection and tracking algorithms[28,29]. Briefly, we applied a feature detection algorithm to identify LAMP1-RFP objects and then fit multiple Gaussians to approximate the center of the objects. Next, we reconstructed individual trajectories for the entire video (i.e., over a 10–20 s period) (Fig. 6a) and applied a moment scaling spectrum (MSS) analysis to determine the motion type of individual trajectories based on the trend of the object's diffusion over time[28]. This approach resulted in a mean of ~200 trajectories extracted per cell and a total of >5000 trajectories per condition analyzed. The majority of the EL trajectories fit within three motion-type categories: confined (blue), free (green), or directed (red), with a much smaller portion of the trajectories determined to be unclassified (magenta) (Fig. 6a, b). Confined trajectories represent ELs that are either static (e.g., confined-immobile) or constrained by barriers to their movement (e.g., confined-mobile). Free ELs move sporadically with frequent changes in direction, probably resulting from intermittent association with microtubule motors (Fig. 6a, b), while directed ELs move directionally, consistent with movement driven by a strong association with kinesin or dynein motors along microtubules (Fig. 6a–c). Free and directed ELs are overall more motile than confined ELs, and thus, for simplicity, we combined free and directed motion types into a single category when quantifying EL motility.

This analysis showed that SNX19 KO resulted in an increased proportion of free plus directed ELs and a decreased proportion of confined ELs (Fig. 6d), indicative of increased EL motility. In contrast, cells transfected with plasmids encoding SNX19-GFP or the hypertethers SNX19$^{\Delta PXA}$-GFP or SNX19$^{1-659}$-GFP exhibited a decreased proportion of free plus directed ELs and an increased proportion of confined ELs relative to GFP-transfected cells

(Fig. 6e, f). Even in the same field of observation (Fig. 6g), ELs having associated SNX19-GFP (indicated by arrows) were less motile than neighboring ELs devoid of SNX19-GFP (no arrows) (Fig. 6g, h).

Further analysis of EL movement in the peripheral cytoplasm of WT U-2 OS cells (only region where individual ER tubules can be resolved) showed that SNX19-GFP-positive, mostly static LAMP1-RFP puncta were associated with BFP-KDEL–labeled ER tubules 100% of the time, whereas SNX19-GFP-negative, motile LAMP1-RFP puncta were associated with BFP-KDEL-labeled ER tubules ~80% of the time (Supplementary Fig. 6a, b). These data indicated that the binding of SNX19-GFP not only decreases the motility but also enhances the alignment of ELs with ER tubules. It's important to point out that the alignment of 80% motile ELs with ER does not necessarily reflect physical tethering between these organelles, but could result from the fact that both ELs and ER are independently attached to the same microtubules in the peripheral cytoplasm[30,31].

We also tested if apilimod treatment, a condition that enhances SNX19-EL association (Fig. 2f, g), affects EL motility. Because PIKFYVE inhibition also causes EL swelling, an effect that alone may lower EL motility due to friction-based constraints, we treated cells acutely (30 min) with apilimod. This duration of treatment was sufficient to cause an increase in SNX19-EL tethering without a gross volumetric gain of ELs (Fig. 6i). Under these conditions, the acute apilimod treatment strongly reduced EL motility (Fig. 6j).

From these experiments, we concluded that ER–EL contacts mediated by SNX19 restrict EL motility.

**SNX14 does not localize to ER–EL contact sites**. The SNX19 homolog SNX14 was previously shown to localize to ER-LD contacts upon treatment of cells with fatty acids[16]. To determine if SNX14 could also localize to ER–EL contacts, we compared the localization of SNX19-GFP and SNX14-mNeonGreen relative to LAMP1-RFP in U-2 OS cells in the absence of added fatty acids. We observed that, like SNX19-GFP, SNX14-mNeonGreen localized to the reticular ER (Fig. 7a). However, whereas SNX19-GFP also localized to puncta containing LAMP1-RFP, SNX14-mNeonGreen did not (Fig. 7a, b). Moreover, treatment with apilimod greatly increased the number of SNX19-GFP–EL contacts, but did not induce the formation of SNX14-mNeonGreen–EL contacts (Fig. 7b, c).

Conversely, we tested whether SNX19 could be recruited to LDs upon treatment with fatty acids. We observed that treatment of cells with oleic acid (OA) for 2 h did cause SNX19-GFP to associate with LDs labeled with GFP-ADRP (Fig. 7d, e) and monodansylpentane (MDH) (Supplementary Fig. 7), as was previously reported for SNX14[16]. Despite this association, 3-color time-lapse imaging of live cells revealed that SNX19-Halo–EL contacts were maintained upon treatment with OA and that SNX19-Halo could independently contact ELs (labeled with LAMP1-RFP) and LDs (labeled with GFP-ADRP) under this condition (Fig. 7f, g).

Taken together, these experiments indicated that SNX19 can contact both ELs and LDs, whereas SNX14 only contacts LDs.

## Discussion
Our studies have thus identified SNX19 as a player in the mechanisms that regulate EL motility and positioning. SNX19 is anchored to the ER by its two N-terminal TM domains and contacts ELs by the interaction of its PX domain with the endolysosomal lipid PI(3)P. These interactions result in the concentration of SNX19 at sites of ER–EL contact. Since the ER is generally less motile than ELs, the net result is a reduction in

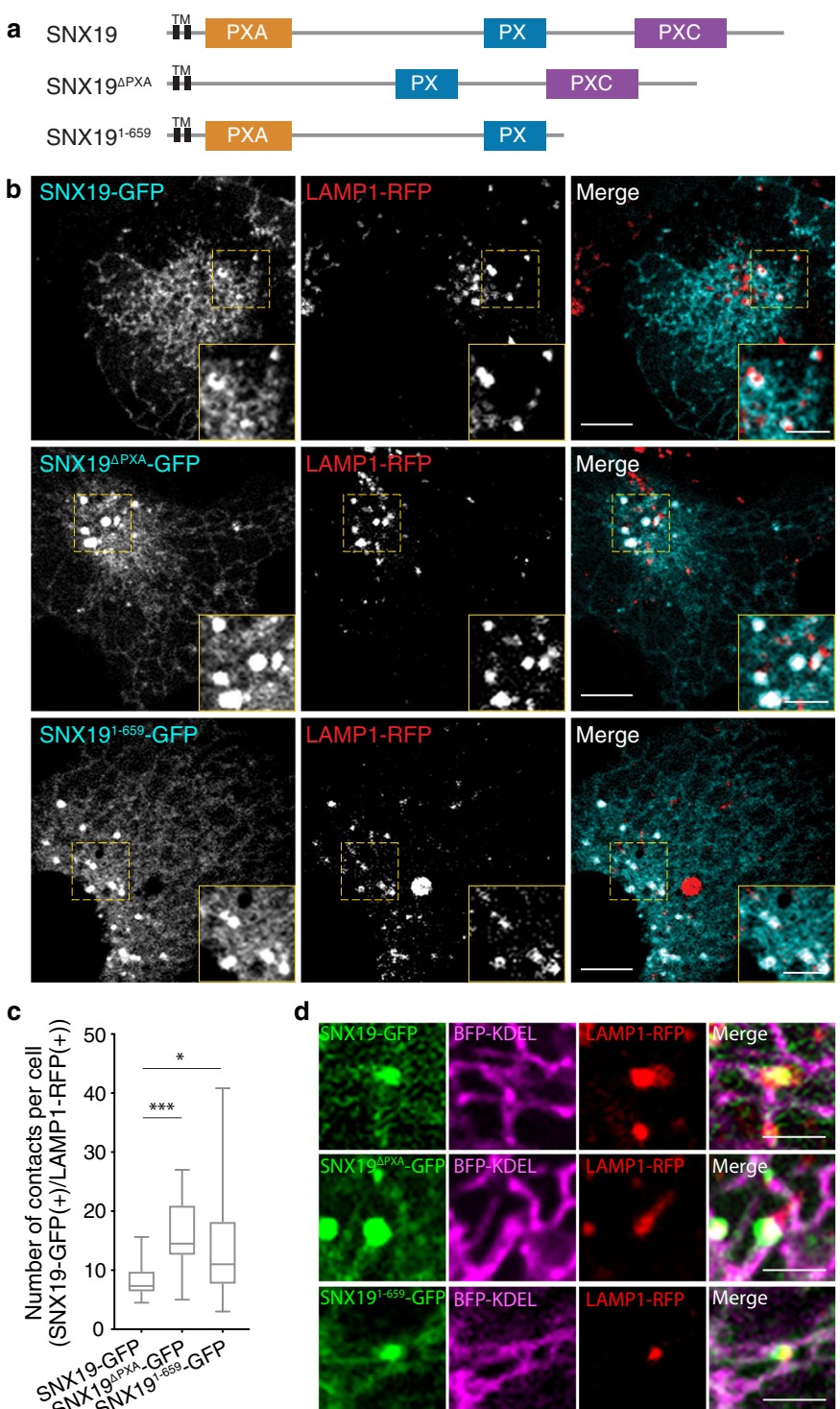

**Fig. 3 Deletion of SNX19 PXA or PXC domains results in hypertethering to endolysosomes. a** Scheme of WT SNX19 and SNX19 mutants lacking the PXA domain or the C-terminal region containing the PXC domain. Scheme not to scale. **b** U-2 OS cells were co-transfected with plasmids encoding GFP-tagged SNX19 constructs depicted in **a** together with LAMP1-RFP, and imaged live by confocal microscopy. Insets are zoomed-in areas of dashed boxes. **c** Quantification of the number of contacts between SNX19-GFP constructs and LAMP1-RFP observed under conditions in **b**. The data are represented as box and whisker plot with the box extending from 25th to 75th percentiles, the horizontal line representing the median, and the whiskers representing the minimum and maximum values. Data is from $n = 3$ independent experiments, where 10–20 cells were quantified per condition, per experiment. Significance was computed by Kruskal–Wallis one-way ANOVA; *$p = 0.0217$, ***$p < 0.001$. Source data are provided as a Source Data file. **d** U-2 OS cells were co-transfected with plasmids encoding BFP-KDEL, LAMP1-RFP, and each of the GFP-tagged SNX19 constructs depicted in **a**, and cells were imaged live by confocal microscopy. Images show close-up SNX19-GFP–EL contacts. Experiment was independently conducted at least three times and was reproducible. Scale bars: 10 μm in **b**, 5 μm insets and in **d**.

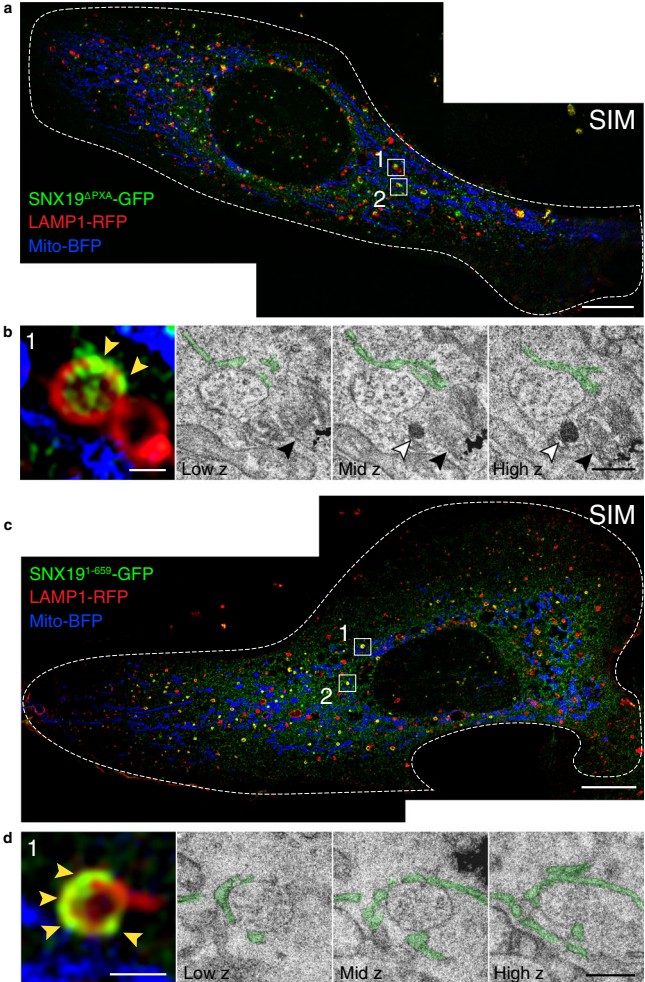

**Fig. 4 CLEM shows localization of SNX19 hypertethers to ER–EL contact sites. a–d** U-2 OS cells co-transfected with plasmids encoding LAMP1-RFP, Mito-BFP, and SNX19$^{\Delta PXA}$-GFP (**a**, **b**) or SNX19$^{1-659}$-GFP (**c**, **d**) were fixed and imaged by structured illumination microscopy (SIM) and transmission EM. **a**, **c** SIM images are stitched and flattened projections of 20 *z* slices each. Boxes 1 and 2 in each image show areas of interest that were further analyzed by EM. **b**, **d** Fluorescence micrographs show enlarged SIM images of contacts in box 1 from **a**, **c**. Yellow arrowheads point at SNX19$^{\Delta PXA}$-GFP (**b**) or SNX19$^{1-659}$-GFP (**d**) enrichment on LAMP1-RFP-positive ELs. The same contacts are shown in EM micrographs at 3 different *z* slices (right). Green shading highlights the ER. Black arrowheads point to a lysosome containing aggregated BSA-gold. White arrowheads point to a LD (**b**). Similar images of box 2 in **a** and **c** are shown in Supplementary Fig. 4a, b. Scale bars: 10 μm in **a** and **c**, and 500 nm in **b** and **d**.

motility of the tethered ELs. Moreover, because the ER is most abundant in the perinuclear region of the cell, the interactions also contribute to a more perinuclear distribution of ELs. The control of EL motility and positioning by SNX19-mediated tethering to the ER likely contributes to the regulation of the various functions that depend on EL distribution within the cytoplasm[5].

**Characteristics of SNX19-mediated ER–EL contacts.** Immuno-fluorescence microscopy and CLEM showed that ER-anchored SNX19 interacts with a range of EL compartments, from early endosomes to lysosomes, with the greatest frequency seen for organelles with characteristics of late endosomes/MVBs (Figs. 1h,

4b, d and Supplementary Figs. 1, 4a). This distribution closely matches that of PI(3)P in the endolysosomal system[19], consistent with the specificity of the SNX19 PX domain for this phosphoinositide (Fig. 2b)[17,18]. These properties are reminiscent of those of protrudin, another protein that is anchored to the ER by three N-terminal hydrophobic domains and that interacts with ELs via a C-terminal PI(3)P-binding FYVE domain[9]. Since the levels of PI(3)P are subject to regulation by nutrients and other stimuli[32–34], the PI(3)P-binding specificity of both SNX19 and protrudin provides a means to control the formation of ER–EL contact sites. These similarities notwithstanding, SNX19 and protrudin differ in other respects. First, whereas SNX19 KO causes dispersal of ELs toward the cell periphery (Fig. 5e, f), protrudin knockdown (KD) causes clustering of ELs in the perinuclear region[9]. These opposite effects might be explained by the ability of protrudin to interact with the small GTPase RAB7 and to transfer the microtubule motor kinesin-1 to the RAB7-effector protein FYCO1 on ELs for transport of ELs toward the cell periphery[9]. Thus, rather than immobilizing ELs in the perinuclear region, protrudin mobilizes them toward the cell periphery. We considered the possibility that SNX19 could exert its effects on EL motility and positioning through interactions with microtubule motors. However, co-immunoprecipitation and pull-down experi1ments failed to detect interactions of SNX19 with kinesin-1, kinesin-3, and dynein (Supplementary Fig. 6c–e). Furthermore, protrudin has an FFAT motif (i.e., two phenylalanines within an acidic tract) that mediates binding to VAP, resulting in a concentration of protrudin at a specific type of ER–EL contact site[35]. In contrast, SNX19 lacks FFAT or similar motifs, and co-expression of VAP-A-Halo with SNX19-GFP or SNX19-GFP hypertethers does not result in accumulation of VAP-A-Halo at SNX19-GFP ER–EL contact sites (Supplementary Fig. 3a–c and Supplementary Movie 2). This property also distinguishes SNX19 from other proteins that localize to ER–EL contact sites by virtue of VAP–FFAT-motif interactions, such as the sterol-binding proteins ORP1L[10], STARD3, and STARD3NL[36], and the phospholipid-transport protein VPS13C[37]. SNX2 is another sorting nexin that localizes to ER–EL contact sites via interactions of its PX domain with PI(3)P and an FFAT motif with VAP[38]. In this case, however, the interactions function to modulate PI(4)P- and WASH-dependent fission of recycling endosomal tubules[38]. Thus, the SNX19-mediated ER–EL contacts identified here differ from those described before, as they are devoid of VAP. Further studies will be needed to determine whether SNX19 interacts with any small GTPases or other components of ER–EL contact sites, and how these interactions are regulated.

**Regulatory role of the SNX19 PXA and PXC domains.** The finding that deletions of SNX19 regions comprising the PXA or PXC domains enhance ER–EL association (Fig. 3) suggests that these regions play a regulatory role. This regulation could involve an intramolecular autoinhibitory interaction, whereby the PXA- and PXC-containing regions prevent PI(3)P binding by the PX domain from the same molecule. Alternatively, the PXA- and PXC-containing regions could inhibit the PX domain of another SNX19 molecule in *trans*. Furthermore, the PXA- and PXC-containing regions could interact with other proteins that, in turn, inhibit ER–EL tethering. Although little is known about the molecular functions of these domains, recent work has shown that the PXA domain of *S. cerevisiae* Mdm1 binds LDs, fatty acids, and the fatty acyl-CoA ligase Faa1[12]. In addition, the PXC domain of human SNX14 binds LDs via an amphipathic helix[16]. The SNX19 PXC domain is also predicted to contain at least two amphipathic helices[39]. Therefore, another possibility is that the PXA and PXC domains of SNX19 regulate the PX domain by binding to other lipids rather than proteins. These lipids could be

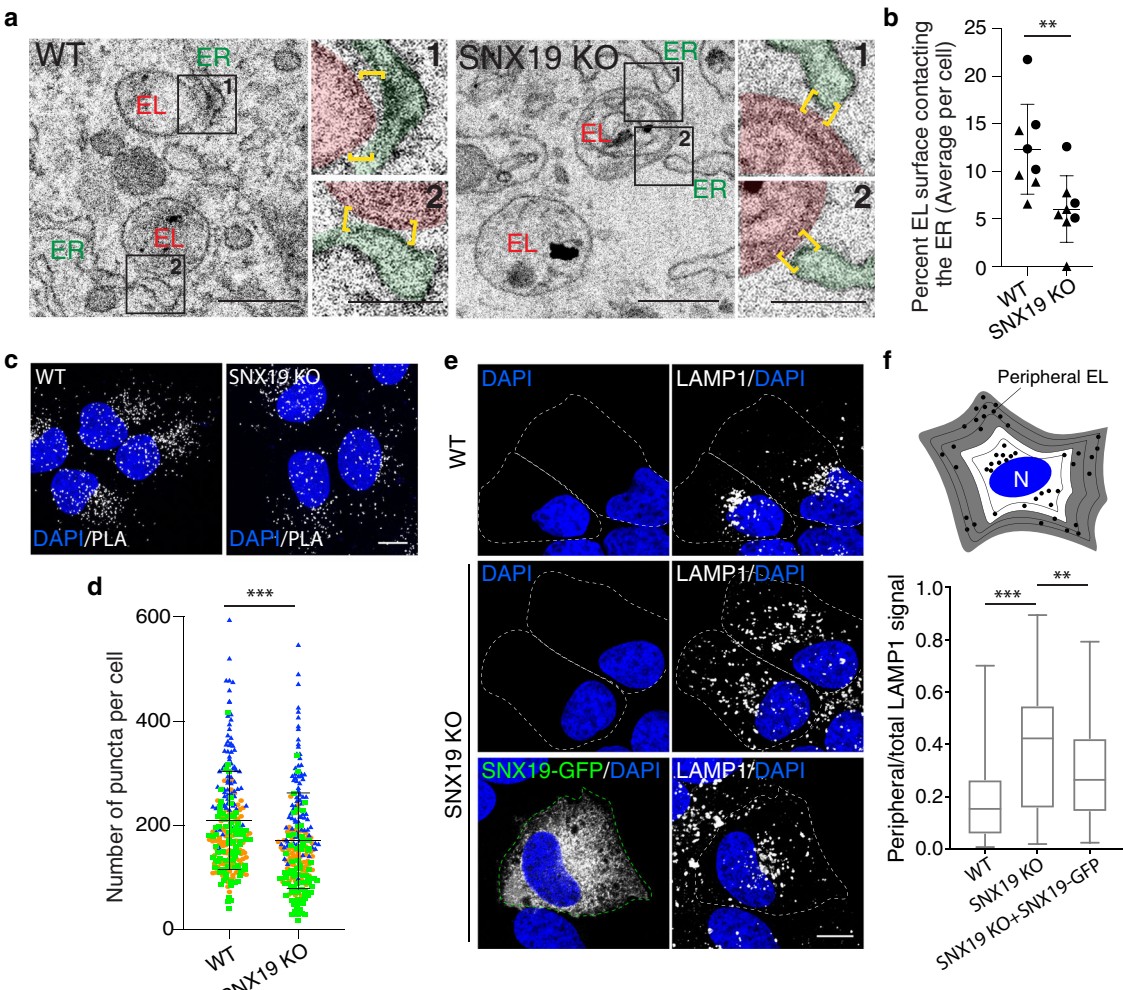

**Fig. 5 Loss of SNX19 reduces overall ER–EL contacts and causes EL dispersal. a** WT or SNX19-KO U-2 OS cells were allowed to endocytose BSA-gold for 24 h to label ELs. Cells were then fixed, embedded in plastic, and subjected to serial sectioning (50-nm sections). Images show single EM sections with representative ELs. Insets are close-up images of ER–EL contacts, where ELs are shaded in red, the ER is shaded in green and yellow brackets define the ER–EL contact sites. **b** Quantification of data in **a**. For each section that captured a gold-containing EL, the perimeter of the EL membrane was measured. The same was done for the portion of the EL membrane that was closely apposed to an ER membrane (i.e., contact length). All EL perimeters and contact lengths across the sections that captured a given EL were summed up, the total contact lengths were divided by the total EL perimeter lengths and multiplied by 100 to compute the percentage of the EL perimeter that contacts the ER. Contacts were quantified in multiple sections from $n = 8$ cells per condition across two independent experiments. Graph shows a line at the mean and SD. Unpaired $t$-test was performed, ** two-tailed $p$-value = 0.009. **c** WT or SNX19-KO U-2 OS cells were fixed, immunostained with rabbit anti-LAMP1 and mouse anti-calnexin antibodies. A proximity ligation assay (PLA) was then performed using anti-mouse and anti-rabbit probes, and detected in the 647-nm channel. The PLA signal is shown in white in two representative confocal micrographs. **d** Quantification of PLA puncta from **c**. A threshold was applied to eliminate background signal and the total number of puncta per cell was computed in three experiments, each consisting of 60–80 cells per condition. Graph shows a line at the mean and SD. Unpaired $t$-test was performed, *** two-tailed $p$-value < 0.001. **e** WT or SNX19-KO U-2 OS cells were fixed, immunostained with anti-LAMP1 antibody, and imaged by confocal microscopy. Bottom row shows a SNX19-KO cell that was transfected with a SNX19-GFP construct. **f** Quantification of peripheral ELs from experiments such as that shown in **e**. Cell outlines were traced and the outline was iteratively shrunk by 2 μm to generate 5 "shells" per cell. The scheme on top depicts combinations of the 3 outermost shells in gray, and 2 innermost shells in white. LAMP1 signal in the gray region (peripheral ELs) was quantified and divided by LAMP1 signal in combined gray and white regions (total ELs). Box and whisker plot represents data from 3 independent experiments where 7–20 cells were analyzed per condition in each experiment. The box extends from 25th to 75th percentiles, the horizontal line represents the median, and the whiskers represent the minimum and maximum values. Statistical significance was calculated with the Mann–Whitney test; *** two-tailed $p$-value < 0.001, ** two-tailed $p$-value = 0.0081. Source data are provided as a Source Data file. Scale bars: 500 nm in **a**, 250 nm insets and 10 μm in **c** and **e**.

on LDs, raising the possibility that SNX19 becomes activated for EL tethering in the proximity of LDs. Alternatively, such lipids could be on the EL membranes themselves. In this regard, it is worth noting that the *S. pombe* protein Pxa1 (not to be confused with the *S. cerevisiae* peroxisomal fatty acyl-CoA transporter Pxa1), whose single defining feature is an N-terminal PXA domain, is involved in protein sorting to the vacuole[40]. This

function suggests a role of the SNX19 PXA domain in direct connection with ELs.

**Functional comparisons between SNX19 and SNX14.** Whereas SNX19 can concentrate at ER–EL contact sites and regulate EL positioning/motility (Figs. 1d–h, 5 and 6), SNX14 lacks these

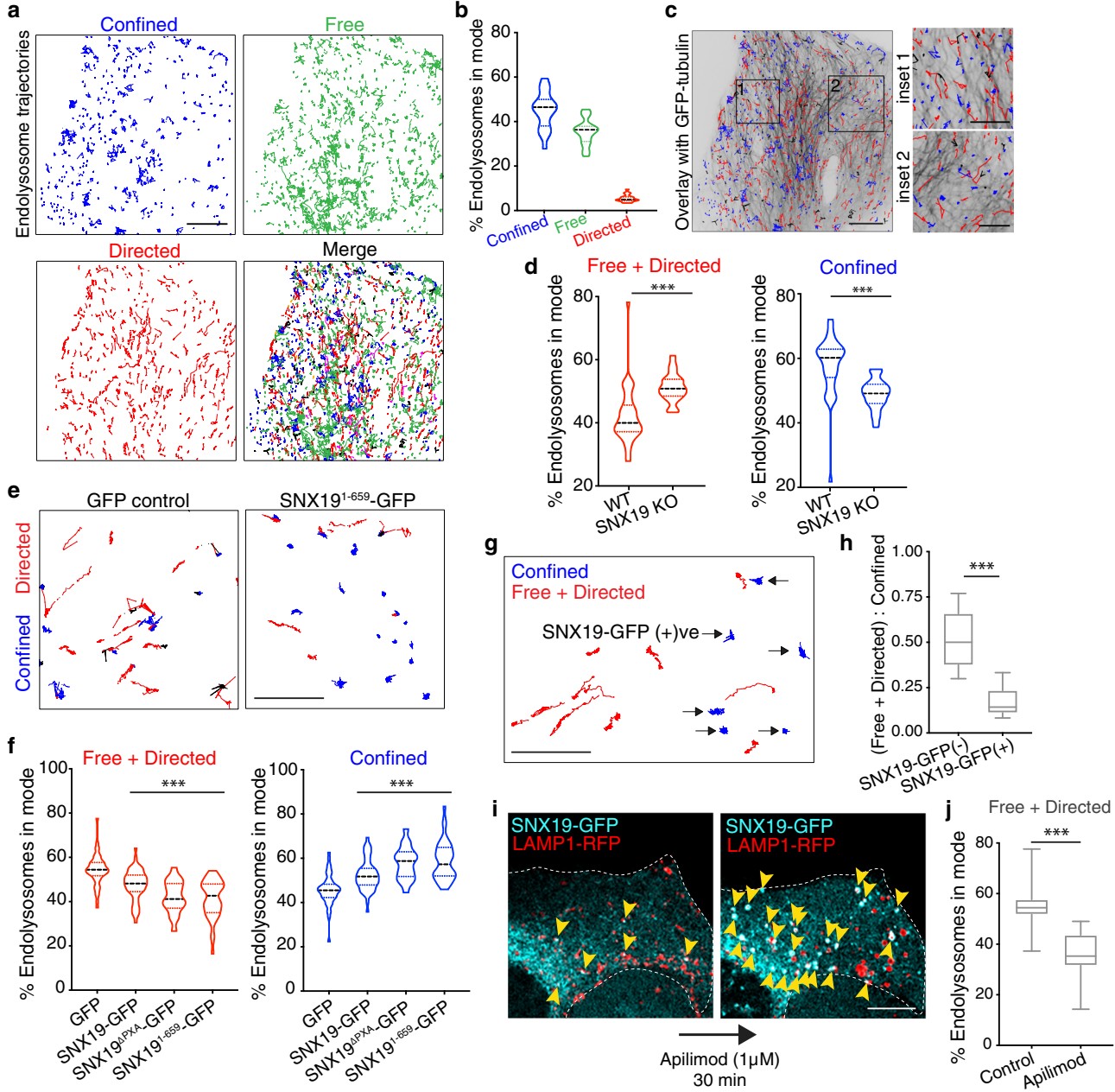

features (Fig. 7a, b). Not even when PI(3)P levels are raised by treatment with apilimod is SNX14 able to interact with ELs (Fig. 7b, c). These functional differences likely stem from the inability of the SNX14 PX domain to bind PI(3)P because of a single amino-acid substitution in the PI(3)P-binding site[17,18]. Likewise, the PX domain of *Drosophila* SNZ is incapable of binding PI(3)P, probably explaining why this protein does not localize to ER–EL contacts but to ER-PM contacts[13]. The PX domain of SNX25 also does not bind PI(3)P, instead binding PS, PI(3,4)$P_2$, PI(4,5)$P_2$, PI(3,5)$P_2$ and PI(3,4,5)$P_3$[17,18]. The SNX13 PX domain shares with the SNX19 PX domain the property of binding PI(3)P[17,18], and indeed a SNX13-GFP construct also co-localizes with LAMP1-RFP at ER-associated foci (Supplementary Fig. 6f), making SNX13 another candidate for mediating ER–EL contacts. These observations highlight the functional diversification of this SNX subfamily in mammals.

An activity that SNX19 and SNX14 do share is their association with LDs upon treatment of cells with OA (Fig. 7d–f and refs. [15,16]). For SNX14, this activity enables its function in lipid metabolism at the ER-LD interface[16]. SNX19 could therefore be involved in the same process. Nevertheless, SNX19 does not seem able to functionally compensate for SNX14 loss, as KO of SNX14 alone in U-2 OS cells causes defects in LD morphology[16]. Further assessment of our SNX19-KO cells will be needed to determine whether SNX19 plays a role in lipid homeostasis and to test if SNX14 and SNX19 have overlapping functions at ER-LD contacts.

**Other functions of SNX19**. SNX19 has been shown to interact with islet antigen-2 (IA-2), a protein tyrosine phosphatase and autoantigen in type I diabetes[41,42] that is expressed in brain neuroendocrine cells, pituitary gland, and pancreas. In pancreatic β cells, SNX19 silencing results in fewer dense-core vesicles (DCVs) and decreased insulin content and secretion[43]. Given that both IA-2 and PI(3)P are present on DCVs[44,45], SNX19 may be recruited to DCVs by coincident detection of these two factors to control DCV abundance. Unlike IA-2, however, SNX19 expression is not restricted to regulated secretory cells, consistent with a more general

**Fig. 6 SNX19 restricts endolysosome motility. a** U-2 OS cells were co-transfected with plasmids encoding LAMP1-RFP and GFP-tubulin, imaged live in the red channel at 10 fps for 1 min, and LAMP1-RFP micrographs were subjected to MSS analysis to identify ELs, build EL tracks and classify their motion types: confined, blue; free, green; directed, red; unclassified, magenta. **b** Quantification of LAMP1-RFP tracks generated by MSS from cells treated and imaged as in **a**, plotted according to the 3 main motion types. Data from 3 independent experiments, each consisting of 7–9 cells, are represented as violin plots with lower and upper quartiles in colored dashed lines and the median as a black dashed line. **c** Overlay of confined (blue) and directed (red) LAMP1-RFP tracks with GFP-tubulin from cell in **a**. **d** WT and SNX19-KO U-2 OS cells were transfected with a plasmid encoding LAMP1-RFP, imaged live for 20 s at 10 fps, subjected to MSS analysis, and EL trajectories were plotted as the proportion of ELs with combined free and directed motion (left), or confined motion (right). Data from 3 independent experiments, consisting of 8–15 cells per condition per experiment, are represented as violin plots with a thick dashed line representing the median and thin dashed lines representing the quartiles. Mann–Whitney test was applied to determine significance; *** two-tailed *p*-value < 0.001. **e** U-2 OS cells were co-transfected with plasmids encoding GFP or SNX19$^{1-659}$-GFP plus LAMP1-RFP, imaged live as described in **d**, and EL tracks were generated by MSS. **f** U-2 OS cells were co-transfected with plasmids encoding GFP, WT SNX19-GFP, SNX19$^{\Delta PXA}$-GFP, or SNX19$^{1-659}$-GFP plus LAMP1-RFP, and EL tracks were plotted as the proportion of ELs with combined free plus directed motion (left) or confined motion (right). Data from 3 independent experiments, consisting of 8–15 cells per condition in each experiment, are represented as violin plots with a thick dashed line representing the median and thin dashed lines representing the quartiles. Statistical significance was analyzed by one-way ANOVA; ***$p$ < 0.001. **g** EL tracks generated by MSS from the SNX19-GFP condition in **f**. Arrows point to ELs that had visible SNX19-GFP accumulated on them. **h** Twelve randomly selected videos across 3 independent experiments from the SNX19-GFP condition in **f** were analyzed for the absence (−) or presence (+) of visible SNX19-GFP accumulation on ELs as in **g**, and those EL tracks were represented as box and whiskers plots of the ratio of combined free plus directed motion to confined motion. The box extends from 25th to 75th percentiles, the horizontal line represents the median, and the whiskers represent the minimum and maximum values. Mann–Whitney test was applied to determine significance; *** two-tailed *p*-value < 0.001. **i** Live-cell imaging of the same U-2 OS cell co-transfected with plasmids encoding SNX19-GFP and LAMP1-RFP before (left) and after (right) acute (30 min) treatment with 1 μM apilimod. Yellow arrowheads point to SNX19-GFP–LAMP1-RFP contacts. **j** U-2 OS cells were co-transfected with plasmids encoding GFP plus LAMP1-RFP, incubated or not for 30 min in the presence of 1 μM apilimod, and imaged live as described in **d**. The proportion of ELs exhibiting combined free plus directed motion types in control (GFP condition from **f**) cells and cells treated with apilimod was plotted. Box and whisker plot of data from 18 cells across 3 experiments where the box extends from 25th to 75th percentiles, the horizontal line represents the median, and the whiskers represent the minimum and maximum values. Statistical significance was analyzed by the Mann–Whitney test, *** two-tailed *p*-value < 0.001. Source data are provided as a Source Data file. Scale bars: 10 μm in **a**, **c**, **i**, 5 μm insets and **e** and **g**.

function in controlling EL motility. SNX19 has been additionally implicated in the partitioning of the dopamine D1 receptor (D1R) to lipid rafts[46] and in chondrogenic differentiation[47]. It is unclear, however, if these functions relate to the ability of SNX19 to tether ELs to the ER.

**Disease connections.** Whereas biallelic mutations in the *SNX14* gene are known to cause autosomal recessive spinocerebellar ataxia 20 (SCAR20), a disorder characterized by cerebellar atrophy, ataxia, and severe intellectual disability[48,49], mutations in the *SNX19* gene have not yet been shown to cause a Mendelian disorder[50–55]. Genetic and epigenetic analyses, however, have linked variations in *SNX19* expression to proliferation and apoptosis in pancreatic β cells[56], coronary heart disease[57], and cancer[58,59]. In addition, extensive analyses have revealed an association of higher levels[51,52] or rare transcript variants of *SNX19*[53] with schizophrenia risk. Among these variants, a prominent risk-associated transcript is predicted to encode a protein lacking the C-terminal PXC domain, a mutation similar to our SNX19$^{1-659}$ hypertether. The expression of higher levels of normal SNX19 or normal levels of PXC-deleted SNX19 would be expected to enhance EL tethering to the ER, decreasing the motility of ELs. This defect, particularly in relevant neuronal types, could contribute to the pathogenesis of this form of schizophrenia. Since SNX19 can also associate with LDs, defective handling of lipids at the ER-LD interface could also underlie the development of this disorder.

**Concluding remarks.** In general, sorting nexin proteins have been shown to play various roles in intracellular cargo sorting and cell signaling[14]. Studies on Mdm1, SNZ and SNX14 expanded these roles to the formation of ER-organelle contacts for the regulation of lipid metabolism[11,13,16]. Our studies on SNX19 now demonstrate another function for a sorting nexin in the regulation of EL motility and positioning by tethering to the ER.

## Methods
All primers used in this study are listed in Supplementary Table 1.

**Plasmid constructs.** The following plasmid constructs were generated in this study:

*SNX19-GFP*: Human SNX19 cDNA was PCR-amplified using primers AS191 and AS318 and inserted into *Eco*RI-digested pEGFP-N1 using Gibson Assembly (E2611L, New England Biolabs, Ipswich, MA, USA).

*SNX19$^{\Delta TM}$-GFP*: cDNA corresponding to amino acids 95-992 of SNX19 was PCR-amplified using primers AS317 and AS318 and inserted into *Eco*RI-digested pEGFP-N1 using Gibson Assembly.

*SNX19-Halo*: The Halo tag cDNA was PCR-amplified using primers AS306 and AS307 and swapped into SNX19-GFP by digesting out the GFP sequence with *Age*I and *Not*I.

*SNX13-GFP*: Human SNX13 cDNA was PCR-amplified using primers AS187 and AS188 and inserted into *Eco*RI-digested pEGFP-N1 using Gibson Assembly.

*SNX19$^{\Delta TM\Delta PXA}$-GFP*: cDNA corresponding to amino acids 272-992 of SNX19 was PCR-amplified using primers AS319 and AS318 and inserted into *Eco*RI-digested pEGFP-N1 using Gibson Assembly.

*SNX19$^{R582Q}$-GFP*: Generated by mutagenesis of SNX19-GFP using Q5 site-directed mutagenesis kit (Cat# E0554S, New England Biolabs, Ipswich, MA, USA) and primers AS298 and AS299.

*GST-PX*: The PX domain (amino acids 532-659) of SNX19 was PCR-amplified using primers AS293 and AS294 and inserted into *Bam*HI/*Xho*I-digested pGEX-6P-3 bacterial expression vector by Gibson Assembly.

*GST-PX$^{R582Q}$*: Generated by PCR-amplification of the PX domain of SNX19$^{R582Q}$-GFP, using the primers AS293 and AS294. The fragment was inserted into *Bam*HI/*Xho*I-digested pGEX-6P-3 bacterial expression vector by Gibson Assembly.

*SNX19$^{1-659}$-GFP*: cDNA was PCR-amplified using the primers AS191 and AS282 and inserted into *Eco*RI-digested pEGFP-N1 using Gibson Assembly.

*SNX19$^{\Delta PXA}$-GFP*: Deletion of amino acids 95-272 from WT SNX19 was carried out by mutagenesis of SNX19-GFP using Q5 site-directed mutagenesis kit (Cat# E0554S, New England Biolabs) and the primers AS302 and AS303.

*GST-LIC1-CT*: cDNA corresponding to the C-terminus (amino acids 389-523) of human dynein light intermediate chain 1 was PCR-amplified using the primers AS12 and AS10 and ligated into *Bam*HI/*Xho*I-digested pGEX-6P-3 bacterial expression vector.

*SNX14-mNeonGreen*: Human SNX14 cDNA was PCR-amplified using primers AS254 and AS255 and swapped into pLEX-PGK-LAMP1-mNeonGreen by digesting out the LAMP1 sequence with *Bam*HI and *Nhe*I.

*VAP-A-Halo*: Rat VAP-A cDNA was PCR-amplified using the primers AS380 and AS381 and swapped into SNX19-Halo vector by digesting out SNX19 with *Hind*III and *Age*I.

All cDNAs generated in this study were sequence-verified prior to use.

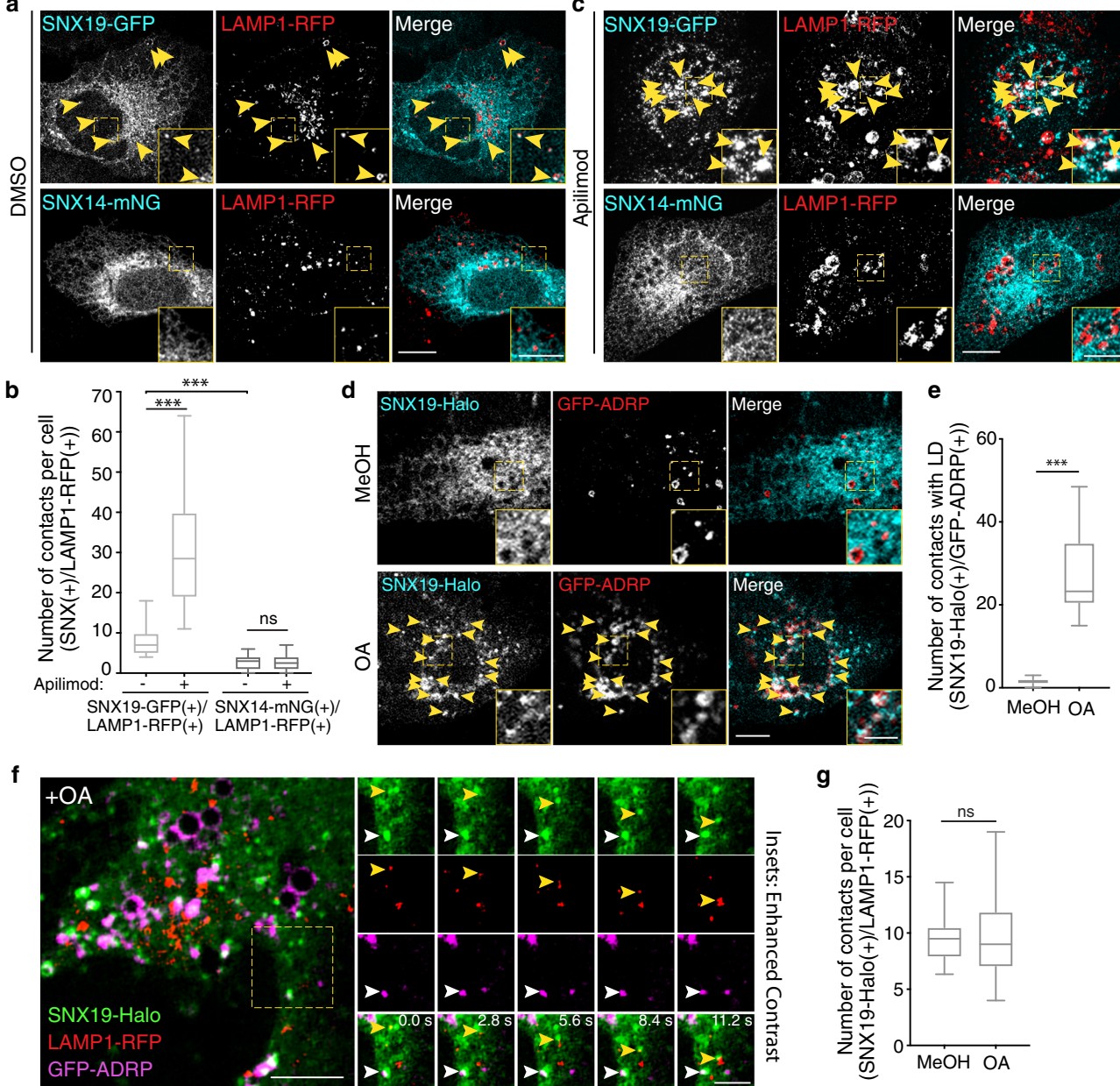

**Fig. 7 SNX14 does not form contacts with endolysosomes. a** U-2 OS cells co-transfected with plasmids encoding LAMP1-RFP and SNX19-GFP or SNX14-mNeonGreen (abbreviated mNG) were treated with DMSO for 2 h, fixed and imaged by confocal microscopy. Arrowheads point to contacts of SNX19-GFP with LAMP1-RFP. Insets are magnified views of the boxed areas. **b** Quantification of contacts in **a** and **c**. Box and whisker plots of data from 20 to 30 cells across 2 independent experiments. The box extends from 25th to 75th percentiles, the horizontal line represents the median, and the whiskers represent the minimum and maximum values. Statistical significance was calculated using the Mann–Whitney test; *** two-tailed *p*-value < 0.001, ns: not significant. **c** U-2 OS cells co-transfected as in **a** were treated with 200 nM apilimod for 2 h, fixed, and imaged by confocal microscopy. Arrowheads point to contacts of SNX19-GFP with LAMP1-RFP. Insets are magnified views of the boxed areas. **d** U-2 OS cells co-transfected with plasmids encoding SNX19-Halo (labeled with Janelia Fluor 646 dye) and the LD marker GFP-ADRP were treated with either methanol (MeOH) as a vehicle control (top panel) or 200 μM oleic acid (OA) (bottom panel) for 2 h, and imaged live by confocal microscopy. Arrowheads point to contacts of SNX19-Halo with LDs. Insets are magnified views of the boxed areas. **e** Quantification of SNX19-Halo–GFP-ADRP contacts under conditions in **d**. Box and whisker plots represent data from 3 independent experiments, 10 cells per condition, per experiment. Statistical significance was analyzed by the Mann–Whitney test; *** two-tailed *p*-value < 0.001. **f** U-2 OS cells co-transfected with plasmids encoding SNX19-Halo (labeled with Janelia Fluor 646 dye), LAMP1-RFP, and GFP-ADRP, were treated with oleic acid (OA) as in **d**, and imaged live by confocal microscopy. Small images are frames from a video with the individual and merged channels cropped from the dashed box over time. Yellow arrows: SNX19-Halo–LAMP1-RFP contacts, White arrows: SNX19-Halo–GFP-ADRP contacts. **g** Quantification of SNX19-Halo–LAMP1-RFP contacts in experiments described in **f** where control cells are those treated with MeOH. Box and whisker plots represent data from 3 independent experiments, 6–18 cells imaged per condition, per experiment. A two-sided Mann–Whitney test was used to analyze statistical significance; ns: not significant. For plots in **b**, **e**, **g**, box extends from 25th to 75th percentiles, the horizontal line represents the median, and the whiskers represent the minimum and maximum values. Source data are provided as a Source Data file. Scale bars: 10 μm, 5 μm insets.

The sources of plasmids encoding other proteins were as follows: LAMP1-RFP was a gift from Walther Mothes (Addgene plasmid #1817), GFP-ADRP was from GeneCopoeia (Germantown, MD, USA), DsRed-Mito was from Clontech (Mountain View, CA, USA), Halo-SEC61B was a kind gift from the Lippincott-Schwartz laboratory[60], 2xFYVE-GFP was a gift from Sergio Grinstein[61], Mito-BFP and BFP-KDEL were gifts from Gia Voeltz (Addgene plasmid #49151 and 49150, respectively), GFP-SKIP was previously made in our lab[62], GFP-RILP was previously described[63].

**Cell culture and transfection**. U-2 OS (Cat# HTB-96, ATCC) and HeLa (Cat# CCL-2, ATCC) cells were grown in DMEM containing 4 mM ʟ-glutamine and 1.5 g/L sodium bicarbonate (Cat# 670087, ThermoFisher Scientific, Waltham MA, USA) supplemented with 10% FBS (Cat# 35-011-CV, Corning, Corning NY, USA) and 100 IU penicillin, 100 μg/mL streptomycin (Cat# 30-002-CI, Corning, Corning NY, USA), and maintained in an incubator at 37 °C, 5% $CO_2$, and 95% relative humidity.

Cells in 4-well live-imaging chambers (Cat# C4-1.5H-N, Cellvis, Mountain View CA, USA) or on 12 mm coverslips (Cat# 12 545 80 P, ThermoFisher Scientific) in 24-well dishes were transfected with 1.4 μg SNX19-GFP or SNX19-Halo and 1 μg LAMP1-RFP or LAMP1-BFP plasmids with 1 μL Lipofectamine 2000 reagent (Cat# 11668027, ThermoFisher Scientific, Waltham, MA, USA). Other plasmid constructs were used at the following amounts: 0.7 μg DsRed-Mito, Mito-BFP, BFP-KDEL or Halo-SEC61B, 0.5 μg GFP-ADRP, 1.4 μg SNX19$^{ΔTM}$-GFP or SNX19$^{R582Q}$-GFP, 1.1 μg SNX19$^{ΔPXA}$-GFP or SNX19$^{1−659}$-GFP, 1 μg SNX19$^{ΔTMΔPXA}$-GFP, 0.1 μg GFP, 1.4 μg SNX13-GFP, or SNX14-mNeonGreen. Briefly, DNA and Lipofectamine were individually mixed in two tubes of 50 μL Opti-MEM (Cat# 31985070, ThermoFisher Scientific) each, incubated at room temperature for 5 min, then the DNA mix was added to the Lipofectamine mix. The 100 μL transfection mixes were allowed to sit for an additional 20 min at room temperature. During this period, the medium on cells was replaced with 500 μL Opti-MEM, and cells were returned to the incubator. After 20 min, transfection mixes were added to the cells dropwise, and cells were returned to the incubator. The medium was changed to 500 μL complete DMEM after 1 h. Transfections for CLEM imaging were conducted the same way as above but scaled to 25 mm coverslips in 6-well dishes. DNA amounts and Lipofectamine reagent were used at 4× the amounts indicated above, each pre-mixed in 250 μL Opti-MEM and combined to produce 500 μL transfection mixes which were added dropwise to cells in 1 mL Opti-MEM after 20 min. Following a 1 h incubation, the medium was changed to 2 mL complete DMEM.

**Generation of CRISPR–Cas9 knock-in and knock-out cells**. The RAB5A-Halo knock-in (KI) HeLa cell line was generated by CRISPR–Cas9 and homology-directed repair (HDR). This was done using transient co-transfection of pX458 plasmid, a gift from Feng Zhang (Addgene plasmid #48138) bearing the guide 5′-GCCAATATTCTTCTTTCTGG, targeting a PAM sequence 60 bp upstream the start codon of *RAB5A* in exon 2, and a donor vector containing the Halo tag coding sequence and left and right homology arms (each ~800 bp), to the *RAB5A* genomic region flanking the sgRNA site for targeted insertion of the Halo Tag at the N terminus of RAB5A. A single point mutation was introduced in the donor vector to destroy the PAM sequence and allow proper insertion of the tag in the genome while active Cas9 was transiently expressed. Single-cell sorting of GFP-positive cells was performed. After two weeks, the same cells were incubated with Janelia Fluor 549 dye[64] (see "Inhibitors and other reagents" section for Halo Tag labeling), rinsed twice with fresh medium, lifted with trypsin, and immediately sorted for GFP-negative/Janelia Fluor 549-positive signal to only recover the cells with the insertion of the Halo Tag. Homo/heterozygotic monoclonal cell lines expressing endogenous RAB5A-Halo were screened by PCR using the primers Rab5 Halo KI F and Rab5 Halo KI R. A clone that was edited on one allele was kept for this study (Supplementary Fig. 1b).

To inactivate the *SNX19* gene in U-2 OS cells, the CRISPR–Cas9 system was used with two sgRNA sequences. Initially, 6 different sgRNAs targeting the first exon of *SNX19* near the start codon were designed using the online tool Benchling (https://www.benchling.com). The guides were cloned into pX458 plasmid and co-transfected into U-2 OS cells in different combinations. Three days later, cells were harvested and genomic DNA was isolated. PCR was conducted using the primers AS407 and AS408 to yield a band of 1.1 kb in WT cells and a short band (100–600 bp shorter depending on the sgRNA combination) in CRISPR-edited cells. The PCR products were run on a 1% agarose gel and those sgRNA combinations that together produced the most prominent short band were used as the final guides for generating KO cells. The 2 final guides (sgRNA1 5′-CCGTTCC AGGAAACTCCAGC and sgRNA2 5′-GCTGAAGTCACCTATACGCG) were co-transfected into U-2 OS cells and 24 h later cells were single-cell sorted into 96-well plates on GFP-positive signal and allowed to grow in complete DMEM for 2–3 weeks. When colonies were visible, cells were transferred to larger wells until a confluent well of a 6-well dish could be harvested and PCR-screened as described above. Those colonies that had undergone non-homologous end joining, and showed only the short PCR band, were kept as "KO" and their PCR products were sequence-verified (Supplementary Fig. 5a, b).

**Antibodies**. Primary antibodies used for immunofluorescence microscopy and immunoblotting, and their concentrations and sources, are as follows: mouse anti-calnexin used at 1:500 (Cat# MAB3126, Millipore Sigma, Burlington, MA, USA), mouse anti-LAMP1 [H4A3] used at 1:500 (DSHB Hybridoma Product H4A3, deposited to the DSHB by J.T. August and J.E.K. Hildreth), rabbit anti-LAMP1 [D2D11] used at 1:500 (Cat# 9091, Cell Signaling Technology, Danvers, MA, USA), mouse anti-EEA1 used at 1:500 (Cat# 610456, BD Biosciences, Franklin Lakes, NJ, USA), rabbit anti-Tomm20 used at 1:1,000 (Cat# ab186734, Abcam, Cambridge, UK), mouse anti-GM130 used at 1:500 (Cat# 610823, BD Biosciences, Franklin Lakes, NJ, USA), goat anti-GST HRP-conjugated antibody used at 1:2,000 (Cat#-GERPN1236, Millipore Sigma), rabbit anti-Rab5 [C8B1] used at 1:500 (Cat# 3547, Cell Signaling Technology), mouse anti-Halo Tag used at 1:1,000 (Cat# G9211, Promega, Madison, WI, USA), rabbit anti-KIF5B used at 1:1,000 (Cat# 167429, Abcam, Cambridge, UK), rabbit anti-KLC2 used at 1:1,000 (Cat# 95881, Abcam, Cambridge, UK), rabbit anti-KIF1B used at 1:1,000 (Cat#A301-055A, Bethyl Laboratories, Montgomery, TX, USA), mouse anti-GFP HRP-conjugated used at 1:5,000 (Cat# 130-091-833, Miltenyi Biotec, Bergisch Gladbach, Germany).

The following secondary antibodies, purchased from ThermoFisher Scientific, Waltham, MA, USA, were used for immunofluorescence microscopy at 1:2,000: donkey anti-mouse IgG Alexa Fluor 488 (Cat# A21202), donkey anti-rabbit IgG Alexa Fluor 488 (Cat# A21206), donkey anti-mouse IgG Alexa Fluor 555 (Cat# A31570), donkey anti-rabbit IgG Alexa Fluor 555 (Cat# A31572), donkey anti-mouse IgG Alexa Fluor 647 (Cat# A31571), donkey anti-rabbit IgG Alexa Fluor 647 (Cat# A31573). The following secondary antibodies, also purchased from ThermoFisher Scientific, were used for immunoblotting at 1:10,000: goat anti-rabbit IgG (H + L) HRP-conjugated (Cat# G-21234), goat anti-mouse IgG (H + L) HRP-conjugated (Cat# 62-6520).

**Inhibitors and other reagents**. To inhibit VPS34, cells were treated with 1 μM SAR405 (Cat# 2716, Axon Medchem, Groningen, The Netherlands) for 2 h prior to imaging. PIKFYVE was inhibited with 200 nM apilimod (Cat# sc-480051, Santa Cruz Biotechnology, Dallas, TX, USA) for 2 h. As a control, cells were treated for 2 h with an equivalent volume of DMSO, the solvent that both inhibitors were reconstituted in. Where indicated, acute treatments were conducted using 1 μM apilimod for 30 min.

To visualize Halo-tagged proteins, cells expressing Halo Tag constructs or the HeLa RAB5A-Halo-KI cells were incubated in a medium containing 500 nM Janelia Fluor 549 or 646 cell-permeable dyes (Cat# 6147 and 6148, Tocris, Minneapolis, MN, USA) for 30 min in a 37 °C, 5% $CO_2$ incubator. The cells were washed once with complete DMEM and imaged in fresh pre-warmed complete DMEM.

To label *bona fide* lysosomes in HeLa RAB5A-Halo-KI cells, cells were allowed to endocytose Alexa Fluor 647 (AF647)-conjugated Dextran (Cat# D22914, ThermoFisher Scientific, Waltham, MA, USA) at 10 μg/mL overnight, followed by a minimum of 2 h chase in Dextran-free complete DMEM the following day prior to imaging.

To induce LD growth, cells were treated with oleic acid (Cat# O1008, Sigma Aldrich, St. Louis, MO, USA) for 2 h at a 200 μM final concentration. Briefly, oleic acid was dissolved in complete DMEM at 60 °C, vortexed, cooled to 37 °C, and added to cells. As a control, some cells were treated with the equivalent volume of methanol, the solvent that the oleic acid was reconstituted in. In some experiments, LDs were visualized by staining with MDH (AUTOdot Autophagy Visualization Dye, Cat# SM1000a, Abcepta, San Diego, CA, USA). This was done by washing live cells twice with warm Opti-MEM and adding 1:1,000 MDH in Opti-MEM, for 15 min, in a cell culture incubator. The cells were washed once with Opti-MEM and cell culture medium was added back to the cells. The cells were imaged live and LDs were visualized in the 405 nm channel.

**Immunostaining**. All immunostaining was conducted at room temperature. Cells were fixed in 4% paraformaldehyde in PBS for 20 min followed by 3 washes in PBS. The cells were then permeabilized in PBS plus 0.1% v/v Triton X-100 for 5 min followed by an additional 2 washes with PBS. Cells were blocked in 0.5% w/v BSA in PBS for 30 min, then incubated for 1 h with primary antibodies in 0.5% w/v BSA-PBS. Cells were washed 3 times with 0.5% w/v BSA-PBS followed by incubation with fluorophore-conjugated secondary antibodies for 40 min, protected from light. Cells were then washed 3 times with 0.5% w/v BSA-PBS followed by 3 washes with PBS alone. Finally, coverslips were mounted onto glass slides with Fluoromount-G containing DAPI (Cat# 00-4959, Invitrogen, Carlsbad, CA, USA) and stored in the dark until imaged. For long-term storage, slides were kept at 4 °C.

**Confocal microscopy**. Fixed- and live-cell imaging were conducted on a laser-scanning confocal LSM880 microscope (Zeiss, Oberkochen, Germany) with a PlanApo ×63 1.4 NA objective and the Definite Focus system. Fixed and mounted cells were imaged at room temperature with no $CO_2$ while live cells were imaged on a 37 °C heated stage in a 5% $CO_2$ chamber.

Live-cell imaging requiring higher acquisition speeds than that obtainable with laser-scanning confocal microscopy was conducted on a spinning-disk confocal Eclipse Ti Microscope (Nikon, Tokyo, Japan), fitted with a PlanApo VC ×60 objective with 1.4 NA and an Evolve 512 EMCCD camera (Photometrics, Tucson,

AZ, USA). Cells were kept in focus using the Perfect Focus system and maintained on a 37 °C heated stage in a 5% $CO_2$ humidified chamber. To achieve the high frame rates (>10 fps) required for EL motility tracking, cells were imaged by excitation with red and green lasers and, after emitted light was passed through a beam-splitter (525 nm/605 nm), simultaneous detection using a dual-camera setup.

**PX-domain purification.** Bacterial expression constructs of GST, GST-PX (PX domain of SNX19) and GST-PX$^{R582Q}$, were transformed into *E. coli* BL21 (DE3) (Cat# C2527I, New England Biolabs, Ipswich, MA, USA). The following day, 3–5 colonies of each were picked to inoculate 30 mL starter cultures of Terrific Broth (TB) medium (recipe at http://cshprotocols.cshlp.org/content/2015/9/pdb.rec085894.full?rss=1) containing 100 μg/mL ampicillin (TB + Amp). The starter cultures were incubated at 37 °C with shaking at 220 rpm overnight. The following day, 1 L of TB + Amp was inoculated with the starter culture and incubated at 37 °C with shaking at 220 rpm until $OD_{600} = 1$. IPTG (Cat# I2481, GoldBio, St. Louis, MO, USA) was added at 1 mM final concentration and the bacteria were incubated at 18 °C with shaking at 220 rpm overnight. Bacteria were pelleted and resuspended in 50 mL ice-cold lysis buffer composed of 50 mM Tris pH 8, 150 mM NaCl, 10% glycerol, 1 mM DTT (Cat# DTT-RO, Roche, Basel, Switzerland), 1 tablet of complete EDTA-free protease inhibitor cocktail (Cat# 11 873 580 001, Roche), 0.5 mg/mL lysozyme (Cat# 89833, ThermoFisher Scientific, Waltham, MA, USA) and 50 μg/mL DNase I (Cat# DN25, Sigma-Aldrich, St. Louis, MO, USA). The slurries were left stirring for 30 min at 4 °C followed by sonication and final clearing by centrifugation at 16,000 × *g* for 45 min at 4 °C. Meanwhile, 1.5 mL glutathione-Sepharose 4B resin (Cat# 17-0756-05, GE Healthcare, Chicago, IL, USA) was prepared for each protein by washing 3 times with buffer (50 mM Tris pH 8, 150 mM NaCl, 1 mM DTT, 10% glycerol) containing 1% v/v Triton X-100, followed by 3 washes with buffer only, each wash at 10 resin volumes. The cleared lysate was added to the washed glutathione-Sepharose 4B resin and allowed to mix end-over-end for 2 h at 4 °C. The resin was washed 3 times with buffer, and the proteins were eluted by incubating the resin for 2 min with 2 mL 50 mM Tris-HCl pH 8 and 10 mM glutathione. The elution step was conducted 4 times to yield 8 mL total elution volume. Samples were aliquoted at 1 mg/mL, flash-frozen in liquid nitrogen, and stored at −80 °C.

**PIP strip immunoblotting.** To analyze phospholipid-binding specificity, the purified GST-PX domain of SNX19, GST-PX$^{R582Q}$ domain mutant, or GST alone were incubated with PIP Strips (Cat# P-6001, Echelon Biosciences, Salt Lake City, UT, USA) according to the supplier's protocol. Briefly, the PIP Strips were blocked overnight at 4 °C in a blocking solution composed of 3% w/v BSA in PBS containing 0.1% v/v Tween 20 (PBS-T) under gentle agitation. The following day, the PIP Strips were incubated with purified GST or GST-PX at 0.5 μg/mL in blocking solution and gently agitated at room temperature for 1 h. The strips were washed 3 times with PBS-T, 5 min per wash, and incubated for an additional hour at room temperature with anti-GST HRP-conjugated antibody (1:2,000) in blocking solution. The strips were washed 3 times with PBS-T, 5 min per wash, and the signal was detected by chemiluminescence using SuperSignal West Femto Maximum Sensitivity Substrate (Cat# 34096, ThermoFisher Scientific).

**Correlative light and electron microscopy (CLEM).** U-2 OS cells were seeded onto 25-mm round, photoetched gridded coverslips (Cat# 72265-50, Electron Microscopy Sciences, Hatfield, PA, USA) and transfected the following day. To facilitate the identification of endolysosomal organelles in the EM images, the cells were allowed to endocytose 6-nm BSA-gold particles (Cat# 25483, Electron Microscopy Sciences, Hatfield, PA, USA) for 24 h. The following day, cells were fixed for CLEM in 2% paraformaldehyde, 0.1% glutaraldehyde in PBS at room temperature for 20 min. Cells were washed twice with PBS and placed in a 25-mm round coverglass chamber filled with PBS for imaging. Regions around cells of interest were mapped with a transmitted and fluorescent light on a Zeiss LSM880 microscope (Plan-Apochromat ×63 1.4 NA objective lens) using a tile and stitch module, to capture grid labels as well as cell positions and fluorescence patterns. SIM imaging of fixed cells was then performed on a Zeiss Elyra PS.1 microscope with a Plan-Apochromat ×63 1.4 NA objective lens at room temperature. Three orientation angles of the excitation grid with five phases each were acquired for each *z* plane. Images were then reconstructed with the SIM module in Zeiss ZEN Black software version 14.0.18.201, using the automatic setting.

After imaging, the cells were placed into a fixative solution composed of 2.5% glutaraldehyde, 2% formaldehyde, 2 mM $CaCl_2$ in 0.1 M sodium cacodylate, pH 7.4, for 10 min at room temperature followed by 45 min on ice. The coverslips were washed 4 times for 5 min each, post-fixed with 2% $OsO_4$ for 2 h in the same buffer at 4 °C, extensively washed with water, stained with 2% uranyl acetate in water, dehydrated through a series of increasing ethanol concentrations (30%, 50%, 70%, 90%, 3 changes of 100%) and embedded in EMBed 812 epoxy resin (Cat# 13940, Electron Microscopy Sciences, Hatfield, PA, USA). After resin polymerization, the coverslip was removed with hydrofluoric acid. Cells previously imaged by light microscopy were identified by their position on the grid. A 1 × 1-mm area containing the cell(s) of interest was cut out using a jeweler saw, mounted on an aluminum holder, and trimmed to 300 × 300 μm. Serial 50–70 nm sections were cut parallel to the plane of the coverslip and mounted on formvar/carbon-coated slot (0.5 × 2 mm) EM grids. Sections were stained with 2% uranyl acetate in 50%

ethanol and imaged in a FEI Tecnai 20 transmission electron microscope operated at 120 kV. Images were recorded on an AMT XR81 wide-field CCD camera. Images were analyzed in Fiji/ImageJ (https://imagej.nih.gov/ij/docs/guide/146.html), and 3D reconstructions were generated in Amira software version 6.5.0 (Thermo Scientific, Waltham, MA, USA).

**Measurement of ER–EL contacts by electron microscopy.** WT or SNX19-KO U-2 OS cells were plated on coverslips and incubated with 6-nm BSA-gold particles (Cat# 25483, Electron Microscopy Sciences, Hatfield, PA, USA) for 24 h. The next day, cells were washed twice in PBS, fixed in 4% paraformaldehyde in PBS at room temperature for 20 min and washed again three times in PBS. Cells were further fixed and processed for EM as described in the "CLEM" section. Eight cells were randomly selected in each condition, and a perinuclear and a peripheral region of interest was imaged for each cell, across 20–30 *z*-sections (~50 nm sections) from the base of the cell, resulting in ~50 images per cell. Each image set was analyzed to first identify ELs by the presence of gold particles. Then, for all the sections that captured a given EL, the perimeter of the EL membrane was traced in Fiji/ImageJ and the length was scored in each section. Within the same sections, the lengths of all ER contacts with the same EL membrane (if present) were scored; if no ER contacts were observed, the length of ER contact was zero. The EL perimeter lengths across all sections in which a given EL is present, were summed up. The same was done for the ER contact lengths for that EL. The ER contact length was then divided by the EL perimeter length and multiplied by 100 to get a percentage of the total EL perimeter that contacts the ER. Typically, between 5 and 10 ELs were quantified per cell in this way; thus, we report the per-cell average of these values (Fig. 5b).

**Proximity ligation assay (PLA).** WT and SNX19-KO U-2 OS cells were plated on coverslips and, the following day, PLA was conducted using a mouse primary antibody to the endogenous ER protein calnexin and a rabbit primary antibody to the endogenous EL protein LAMP1 (antibody information in "Antibodies" section), followed by mouse and rabbit PLA probes from the DuoLink In Situ Red Starter Kit Mouse/Rabbit (Cat# DUO92101, Sigma-Aldrich, St. Louis, MO, USA) and detection with DuoLink In Situ Detection Reagents FarRed (Cat# DUO92103, Sigma-Aldrich). The PLA was conducted as per the manufacturer's instructions and coverslips were mounted with Fluoromount-G mounting medium containing DAPI (Cat# 00-4959-52, ThermoFisher, Waltham, MA, USA). Cells were imaged by confocal microscopy in the far red (647 nm) and DAPI (405 nm) channels.

A threshold was applied in the far red channel in Fiji/ImageJ to all acquired images to eliminate background, and all images were converted to binary images. The PLA signal was selected and the Watershed feature was applied to separate any signal dots that were apparently touching. Dots that were at least 5 pixels in size were counted and the total number of dots per cell (individual cells identified by DAPI nuclear stain) was measured across 3 experiments for each of the two conditions.

**Measurement of endolysosomal peripheral dispersal.** To quantify the peripheral distribution of LAMP1 structures, cells with a relatively round shape were chosen for the analysis, as narrow, elongated cells could not be quantified accurately by this method. These requirements were pre-determined and applied to all conditions. Cell outlines were traced in Fiji/ImageJ by briefly boosting the LAMP1 signal such that cytosolic fluorescence could be visualized. The LAMP1 signal was returned to original settings, a threshold was applied to eliminate the background and the total cellular LAMP1 signal was measured. The cell outline was iteratively shrunken by 2 μm to produce a total of 5 shells, shell 1 being the perinuclear region of the cell and shell 5 being the outermost edge of the cell. The LAMP1 signal within each shell was measured. Since WT cells contained the largest fraction of their lysosomes within shells 1 and 2 (represented by the white region in cell scheme in Fig. 5f), we computed the peripheral/total LAMP1 signal as the fraction of LAMP1 signal present in outer shells 3–5 (represented by gray region in the scheme in Fig. 5f) relative to total LAMP1 signal.

**Endolysosome motility tracking.** WT or SNX19-KO U-2 OS cells expressing LAMP1-RFP with or without GFP control, SNX19-GFP, SNX19$^{ΔPXA}$-GFP or SNX19$^{1–659}$-GFP, were recorded for 10–30 s as described in the "Confocal microscopy" section. To determine motion types of individual ELs, recorded LAMP1-RFP data were subjected to Moment Scaling Spectrum (MSS) analysis as previously described[28] (software is publicly available at https://www.utsouthwestern.edu/labs/jaqaman/software/). Endolysosomes exhibited three major motion types: confined, free, or directed. Free and directed tracks were combined when reporting EL motility as they both represent ELs that are overall more motile than confined.

**ER–EL dynamic associations in live-cell imaging experiments.** U-2 OS cells co-transfected with plasmids encoding SNX19-GFP, LAMP1-RFP, and the ER marker BFP-KDEL were imaged live by confocal fluorescence microscopy at a frame rate of 2.4 s. Fiji/ImageJ was used to apply a threshold in each channel to the time-lapse recordings. SNX19-positive LAMP1-labeled organelles outside of the perinuclear area (where the ER signal cannot be resolved) were followed for 10 consecutive

frames. On these time scales, SNX19-positive ELs were usually static. A mask of the EL was generated in each frame by manual tracing of the LAMP1-RFP signal. Then, the LAMP1-RFP masks were overlaid onto the BFP-KDEL signal and the presence of BFP-KDEL within the mask was scored for each frame. The same analysis was done for randomly selected surrounding motile ELs that did not have SNX19 accumulation on them. For each EL type (SNX19-positive, static vs SNX19-negative, motile), the number of frames with BFP-KDEL signal within the LAMP1-RFP mask were divided by 10 (total frames), multiplied by 100, and represented as a percentage of time that the EL is associated with the ER (Supplementary Fig. 6a, b).

**SNX19 co-immunoprecipitation.** To test whether SNX19 interacts with kinesin-1, GFP-Trap (Cat# gtma-20, ChromoTek, Planegg-Martinsried, Germany) was used for co-immunoprecipitation. HEK293T (Cat# 632180, Takara Bio. Inc., Kusatsu, Shiga, Japan) cells were plated at $2.5 \times 10^6$ cells per dish in $3 \times 10$-cm dishes, in 12 mL total medium. The following day, plasmids encoding SNX19-GFP, the negative control GFP or the positive control GFP-SKIP were transfected. Briefly, three tubes with 1.5 mL Opti-MEM were prepared, and 15 µg SNX19-GFP, 0.1 µg GFP, or 3.5 µg GFP-SKIP plasmid DNA were added. In three separate tubes of 1.5 mL Opti-MEM, 25 µL Lipofectamine 2000 reagent was added to each. Each DNA mix was added to a Lipofectamine mix and incubated at room temperature for 20 min. Each 3 mL transfection mix was then added dropwise to one dish of cells. After 24 h, cells were placed on ice, scraped, transferred to 15-mL tubes, and centrifuged at $700 \times g$, for 3 min at 4 °C. The cells were washed three times in ice-cold PBS. Cell lysis and co-immunoprecipitation were carried out as per GFP-Trap manufacturer's protocol. Following final washes, the GFP-Trap magnetic beads were resuspended in 50 µL 2× Laemmli sample buffer and heated at 95 °C for 10 min. Samples were equally loaded on SDS-PAGE followed by immunoblotting to detect endogenous KIF5B and KLC2. The same procedure was conducted to test interaction between SNX19-GFP and kinesin-3 using GFP as a negative control.

**Dynein pulldowns.** To test whether SNX19 interacts with the dynein subunit LIC1, GST, or GST-LIC1-CT (C-terminus of LIC1) were purified as described in the PX domain purification section. However, after final elution, the samples were further purified by gel filtration in 10 mM Tris-HCl (pH 7) buffer with 50 mM NaCl, 2 mM MgCl₂ and 2 mM TCEP on an AKTA FPLC system (GE Healthcare, Chicago, IL, USA) using a HiLoad 16/600 Superdex 200 pg column (GE Healthcare). Fractions containing purified protein were mixed, aliquoted, flash-frozen in liquid nitrogen and stored at −80 °C.

HEK293T cells were transfected with either 1 µg GFP-RILP (positive control) or 15 µg SNX19-GFP as described in the "SNX19 co-immunoprecipitation" section. After washing the cells in ice-cold PBS, cells were extracted with 1 mL ice-cold lysis buffer (50 mM Tris-HCl, pH 7.4, 150 mM NaCl, 1% Triton X-100, complete EDTA-free protease inhibitor cocktail (Cat# 11 873 580 001, Roche)) and rotation for 30 min at 4 °C with occasional pipetting. Lysates were centrifuged at $16,000 \times g$ for 10 min at 4 °C. The lysates were additionally pre-cleared by end-over-end mixing at 4 °C for 1 h with 100 µL glutathione resin (GE Healthcare) that had been pre-washed three times with lysis buffer.

On the same day, the purified GST or GST-LIC1-CT were re-bound to glutathione beads to be used in pulldowns with GFP-RILP and SNX19-GFP lysates. This was done by thawing the GST or GST-LIC1-CT at 4 °C and centrifuging at maximum speed for 30 min at 4 °C to spin out aggregates. Meanwhile, glutathione resin was prepared by setting up 4 tubes with 50 µL resin each and washing three times in 10 mM Tris-HCl (pH 7) buffer with 50 mM NaCl, 2 mM MgCl₂, and 2 mM TCEP (referred to as LIC1 buffer). GST and GST-LIC1-CT were diluted to 1 µg/mL in LIC1 buffer supplemented with a quarter-tablet protease inhibitors (Cat# 11 873 580 001, Roche) (referred to as LIC1 buffer + PI). The diluted GST was added to 2 tubes of glutathione resin (1 mL per tube) and GST-LIC1-CT was added to the other 2 tubes (1 mL per tube). Samples were allowed to rotate at 4 °C for 1 h. The beads were washed three times in LIC1 buffer + PI and the pre-cleared GFP-RILP lysates were divided 50:50 into one GST-glutathione tube and one GST-LIC1-CT-glutathione tube. The same was done with SNX19-GFP lysates. Samples were rotated for 1 h at 4 °C, then washed three times in LIC1 buffer + PI. The resin was resuspended in 50 µL 2× Laemmli sample buffer and heated at 95 °C for 10 min. Samples were equally loaded on SDS-PAGE followed by immunoblotting to detect the GFP- and GST-tagged proteins.

**Quantification and statistics.** To quantify the number of SNX19-positive–LAMP1-positive contacts under a given condition, Fiji/ImageJ was used to apply a threshold in the SNX19 channel to time-lapse recordings; this was done to visualize only SNX19 puncta of higher fluorescence intensity (SNX19 accumulations) than that of its basal ER-resident reticular signal. These higher intensity SNX19 puncta were overlaid with the LAMP1 signal, and puncta that appeared together for at least 10 s were classified as a true contact. A similar analysis was applied to SNX19-positive–ADRP-positive contacts. For experiments comparing SNX19-positive and SNX14-positive EL contacts (Fig. 7a–c), fixed cells were imaged as z stacks. Cells were analyzed by zooming in and thresholding the image as described above to identify higher intensity SNX19 or SNX14 puncta (higher

than the local ER-resident reticular signal) that co-localized with LAMP1-RFP structures. A true contact could usually be observed in two consecutive z slices.

Data were analyzed in Prism version 8.3.1 (GraphPad Software, San Diego, CA, USA). Box-and-whiskers plots show min to max bars and a line at the median. The number of experiments conducted was used as $n$, unless otherwise indicated. Statistical significance was computed by the test indicated in each figure legend. $p < 0.05$ (*), $p < 0.01$ (**), $p < 0.001$ (***).

**Reporting summary.** Further information on research design will be made available in the Nature Research Reporting Summary linked to this article.

## Data availability
All other data supporting the findings of this study will be available from the corresponding author upon reasonable request. Source data are provided with this paper.

## Code availability
The custom Fiji macro for semi-automation of shell analysis used to measure EL positioning is available at: https://github.com/AmraSaric/LysoPosMacro[65]. Tracking software used to measure EL motility was previously described and made publicly available by the authors (https://www.utsouthwestern.edu/labs/jaqaman/software/)[28]. Amphipathic helix predictions for the PXC domain of SNX19 were conducted using the open-source server HeliQuest (https://heliquest.ipmc.cnrs.fr)[39].

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

## Acknowledgements

We thank Xiaolin Zhu for expert technical assistance, Sergio Grinstein, Walther Mothes, Gia Voeltz, Jennifer Lippincott-Schwartz, and Feng Zhang for kind gifts of reagents, Raffaella De Pace for initial advice on the project, Matthias Machner for conducting a search of FFAT motifs within SNX19, Alma Becic for advice on data presentation, Tal Keren-Kaplan and Rafael Mattera for critical review of the manuscript, and other members of the Bonifacino laboratory for helpful discussions.

## Author contributions

A.S., C.M.G., and J.S.B. conceived the project. A.S. and J.S.B. designed experiments and wrote the manuscript. A.S. conducted experiments, analyzed data, and prepared figures. S.A.F. conducted motility tracking of endolysosomes and contributed to manuscript preparation. C.D.W. conducted SIM imaging and helped design the CLEM experiments. M.J. conducted electron microscopy. A.S. and C.D.W. generated EM reconstructions with help from D.C.G. M.S.F. helped with assays to test PX-PIP binding and with design regarding CRISPR–Cas9 editing. C.M.G. and D.C.G. generated RAB5A-HaloTag knock-in HeLa cells.

## Funding

 This project was funded by the Intramural Program of NICHD, NIH (project # ZIA HD001607).

## Competing interests

The authors declare no competing interests.
