## [Peer Review File · Nature Communications]

REVIEWER COMMENTS

Reviewer #1 (Remarks to the Author):

This manuscript is timely, novel, of interest to a wide audience and well written/presented. The study elegantly demonstrates recruitment of sorting nexin SNX19 to ER-endosome contact sites through interaction of its PX domain with endosomal PI3P. The authors go on to demonstrate a role for SNX19 in retention of late endocytic organelles in the perinuclear region, which they attribute to SNX19-dependent contact sites. While this explanation is certainly plausible, I don't feel it has been adequately demonstrated and I would have liked more mechanistic insight into how SNX19 expression regulates endosomal positioning. The suggested endosome retention in the ER-rich perinuclear region simply by virtue of increased SNX19-mediated ER contact could be a contributing factor, but given the transient nature of most membrane contact sites, it seems unlikely that physical contact with ER alone would be sufficient to govern endosome motility. Given that other sorting nexins have been shown to couple directly or indirectly to dynein, an obvious alternative would be that SNX19 at contact sites might mediate recruitment of dynein to endosomes, but this is not explored. I have a couple of additional concerns:

- It is claimed that since SNX19 localises to both ER and endosomes, it regulates ER-endosome contact sites. The authors have generated SNX19 knock-out cells, in which endosome distribution is altered but fail to show any effect on ER-endosome contact sites. To conclude that loss of ER contact underlies the peripheral re-distribution of endosomes, an associated loss of ER-endosome contact site needs to be demonstrated.
- SNX19 colocalisation with LAMP1 is used throughout as a measure of contact sites. To confirm the presence of ER at these sites (as opposed to the all be it quite remote possibility of an endosomal population of SNX19), an additional ER marker should be used. That VAP is excluded from these sites reiterates the need for another ER marker.

I also have the following comments:

1. P4 line 113, not really regulating EL positioning and motility, but maybe interaction of retromer subunit SNX2 with VAP at ER contact sites with endosomal budding sites (Dong et al, 2016) should be mentioned here?
2. P6, Figure 1: would like to see confirmation that the puncta are contact sites by EM, or at the very least by co-staining with an alternative ER marker.
3. Colocalisation between RAB5 (early endosomes) and LAMP1 (late endosomes and lysosomes) is surprising and the duration of the dextran pulse-chase should be given in the legend and/or text.
4. Fig 2: Would be nice to show (absence of) purified mutant (R582) PX interaction with PI3P on PIP strip
5. Fig 3: Again, need to show an additional ER marker.
6. P8, line 234: I don't think it has been established that SNX19 mediates ER-EL contacts. Fig 4 indicates enrichment at the contact but to say it actively mediates the contact, you need to show that ER-EL contacts are reduced/increased on SNX19 loss/overexpression respectively.
7. Fig 5: The effect of SNX19 on endosome distribution is striking but to link it better to contact sites, reduced contacts in the knockdown cells / increased contact in the SNX19-GFP rescue cells needs to be demonstrated.
8. Fig 7: It would be nice to see SNX14 staining with ELs and LDs here as well. Also I find it hard to see the blue-green colocalisation.
9. P12, line 367-373: I couldn't see any evidence of lipid droplets at SNX19-positive contacts in the CLEM images, but are the small electron-dense organelles lysosomes (esp evident in extended data

fig 3)? I'm not sure why they would be so small - do these structures contain BSA-gold (hard to see from the pdf)? If so, these images may suggest a role for SNX19 in the regulation of endosome-lysosome fusion?

10. P13, SNX13: Maybe beyond the scope, but it would be nice to show if SNX13 does localise to ER-endosome contacts – would further strengthen the importance of the PI3P-binding of SNX19 in its role at EL contact sites if it does.

Reviewer #2 (Remarks to the Author):

In this manuscript Saric and colleagues report that SNX19, one of the mammalian orthologues of Mdm1/SNZ, localizes to endoplasmic reticulum (ER) in contact with endolysosomes (EL). They show that the PX domain of SNX19 binds PI3P and inhibition of PI3P synthesis decrease the co-localization of SNX19 with LAMP1-positive EL, while the increase in PI3P increases this co-localization. They also found that deletion of the SNX19 PXA and PXC domains increase the occurrence of ER-EL contacts, suggesting that these domains play a regulatory role. Some hypotheses are discussed to explain their putative role, however the mechanisms underlying such role are not addressed in this study. The most interesting finding of this study is that SNX19 regulates EL positioning and motility, representing the first example of a SNX with this function. Knock-out of SNX19 results in dispersion of EL and in the increase in mobile (here defined as “free” and “plus directed”) vs “confined” EL. Yet, this function and the possible molecular mechanism involved are not further investigated. The authors conclude showing that, upon oleic acid treatment, SNX19 can also localize in proximity to lipid droplets (LDs), and hypothesize that SNX19 might constrain EL motility to position EL in proximity to LDs to maintain lipid homeostasis. However, the existence of a tripartite ER-EL-LD association mediated by SNX19, despite being potentially very interesting, is not convincingly supported by the preliminary low-resolution microscopy analysis presented here.

Major points:

1. The authors show that SNX19 localizes to ER elements closely associated to EL by live cell confocal imaging and also by SIM coupled with CLEM (in the specific case of the PXA deletion mutant). However, to prove with no-doubt that SNX19 is really a tether of ER and EL it should be shown that SNX19 is indeed localized at sites of apposition between the membranes of such organelles (i.e. by APEX-EM, immuno-EM, etc) and that its overexpression or depletion regulates the extent of ER-EL contacts.

For instance, is there a decrease in the extent of ER-EL contact sites upon SNX19 knockdown? Or an increase upon its overexpression?

2. The authors show a novel role (among other SNXs) of SNX19 in restricting endolysosome motility. However, how Snx19 regulates the motility of endosomes remains unclear.

-Is the effect on endolysosome motility directly mediated by ER tethering of EL? The authors show that the ER morphology and distribution in SNX19 KO cells is not affected. How this reconciliates with the EL motility defect? Are the “free” or “directed” endosomes in Snx19 KO cells still in contact with ER?

One possible and easiest way to address this point could be to co-express fluorescently tagged LAMP1 and an ER marker to follow their dynamics and association in SNX19 KO cells.

- Is SNX19 interacting with any rab or kinesin regulating endosomal movement?
- Does SNX19 regulate EL lipid composition and consequently recruitment of motor or other effector proteins?

3. The last figure (Fig 7) of the manuscript compares SNX19 and SNX14 localization, showing that SNX14 does not form contacts with endolysosomes. I find this comparison very pertinent, however I would move it ahead in the manuscript (linked to Figure1 or in Supplementary material).

In this figure the authors also show that SNX19 can additionally localize to ER elements in proximity to LDs, as indicated by the partial overlap of SNX19 and GFP-ADRP (a LD surface protein)(Fig7d). However, the image shown is not very convincing, and the shape of LDs not easy to see. I would suggest using in parallel LD markers such as BODIPY or LTox . Electron microscopy analysis should also be performed to confirm that these structures are indeed ER-LD contact sites and to quantify their extent vs the ER-EL contacts.

In the same figure (Fig 7f) the authors also try to prove the existence of a tripartite contact site association ER-EL-LD mediated by SNX19, but is too difficult to see this association in the images shown, probably given the resolution limits of the approach used and also because the inset images are too small, the individual channel of the entire cell are not shown and quantifications of the occurrence and extent of this tree-way organelle association are lacking (i.e. how many of the ER-EL contacts are also associated to LDs?). To prove the existence of such tripartite association a deeper morphological characterization would be needed, and their functional relevance should be addressed, but I am not sure that this is within the scope of this manuscript.

Minor points:

- Some images seem to be overexposed (e.g. in Figures 1b, 1c, 3b, 7d)
- How the authors explain why the TM deletion mutant of SNX19 (Figure 1c) is not binding, at least in part, the endolysosomes (as the PX domain of SNX19 has been shown to bind PI3P)? If PXA domain inhibits PI3P binding, maybe a doubly-deleted mutant (SNX19 Δ PXA Δ TM) would bind the endolysosomes?
- In Figure 4b1 Snx19 Δ PXA fluorescence seems to be also inside endosomes. Is this area of interest an individual focal plane?
- Method section: How do the authors analyze the colocalization of SNX19 and LAMP1 in fixed cells? How is the cell surface visualized in the EL peripheral dispersal measurements?
- Some typo: Page 7, line 202: "SNX19" instead of "SNAX19"; s23 line 737: "significance" instead of "ignificance"

Reviewer #3 (Remarks to the Author):

Review Saric et al., 2020, Nat. Comm.

The authors characterize the unconventional sorting Nexin SNX19, which harbors two transmembrane domains in addition to the hallmark PX domain. The authors show that

overexpressed SNX19 localizes to ER-EndoLysosome (ER-EL) contact sites, where it restricts EL mobility, probably through tethering the ELs to the less mobile ER. Overall, the study is well written and the data are of high technical quality. The assays are convincing and I largely agree with the authors' conclusions.

I have only one major concern regarding the interpretation of data. Throughout the study, the authors simply assume that SNX19 mediates ER-EL tethering. While I also think that this is the case, this is not really proven by any experiment. SNX19 could just localize to ER-EL contacts which are established by other proteins such as Protrudin and RAB7. It would be great if the authors could show that ER-EL contact sites are altered in their SNX19 KO cells. In theory, there should be less or weakened (more transient?) contact sites. Conversely, overexpression of SNX19 should induce ER-EL contact sites. This could be addressed by FRET analysis between Calnexin and LAMP1, just as an example. Any other assay that can visualize or quantify ER-EL contact sites could also be used instead of FRET analysis.

Overall, I am supportive of publication in Nature Communications as I think that this is an important and interesting study that will garner a lot of interest from the cell biological community. However, the authors should really show that SNX19 does indeed mediate and/or regulate ER-EL contacts. In addition, I have listed some more minor points below.

Additional points:

-Could the authors co-stain SNX19-GFP with some endogenous markers such as EEA1, LAMP1, RAB7? This would corroborate the localization data.

-All the data on SNX19 localization to ELs is derived from overexpressed protein. In the absence of an antibody, can the authors at least overexpress SNX19-GFP (or other tag) at the lowest possible levels in their SNX19 KO cell line and demonstrate that the SNX19-LAMP1 colocalization can still be observed?

-Figure 4: Why did the authors use a mitochondria marker in this figure? To show that SNX19 does not establish mitochondria contacts? The authors should explain why they used a mitochondria marker.

Responses to Reviewers

Reviewer #1:

This manuscript is timely, novel, of interest to a wide audience and well written/presented. The study elegantly demonstrates recruitment of sorting nexin SNX19 to ER-endosome contact sites through interaction of its PX domain with endosomal PI3P. The authors go on to demonstrate a role for SNX19 in retention of late endocytic organelles in the perinuclear region, which they attribute to SNX19-dependent contact sites. While this explanation is certainly plausible, I don't feel it has been adequately demonstrated and I would have liked more mechanistic insight into how SNX19 expression regulates endosomal positioning. The suggested endosome retention in the ER-rich perinuclear region simply by virtue of increased SNX19-mediated ER contact could be a contributing factor, but given the transient nature of most membrane contact sites, it seems unlikely that physical contact with ER alone would be sufficient to govern endosome motility. Given that other sorting nexins have been shown to couple directly or indirectly to dynein, an obvious alternative would be that SNX19 at contact sites might mediate recruitment of dynein to endosomes, but this is not explored.

We thank this reviewer for finding our manuscript “*timely, novel, of interest to a wide audience and well written/presented*”.

In response to this reviewer's introductory comments, we have now tested for physical interaction between SNX19 and dynein by pulldown experiments. This was done by incubating GST-tagged C-terminal domain of dynein light intermediate chain 1 (GST-LIC1-CT), which is known to interact with many adaptor proteins^{1,2}, or GST alone as a negative control, with cell lysates expressing SNX19-GFP or the known dynein adaptor GFP-RILP as a positive control. The results of these experiments showed that GST-LIC1-CT pulled down GFP-RILP but not SNX19-GFP (**new Extended Data Figure 6e**). These findings make it unlikely that SNX19 functions as an adaptor of endolysosomes (ELs) to dynein. These observations are now mentioned in the Discussion.

I have a couple of additional concerns:

- *It is claimed that since SNX19 localises to both ER and endosomes, it regulates ER-endosome contact sites. The authors have generated SNX19 knock-out cells, in which endosome distribution is altered but fail to show any effect on ER-endosome contact sites. To conclude that loss of ER contact underlies the peripheral re-distribution of endosomes, an associated loss of ER-endosome contact site needs to be demonstrated.*

The reviewer brings up a very important point. We have addressed this comment using a couple of approaches that support a contribution of SNX19 to overall ER-EL contacts.

First, we performed transmission electron microscopy of WT and SNX19-KO cells and quantification of EL-ER contacts on the micrographs (**new Figure 5a,b**). These analyses consisted of allowing live cells to endocytose BSA-gold for 24 h to label ELs, after which cells were fixed, embedded in plastic and subjected to serial sectioning (50-nm sections). For each section that captured a gold-containing EL, the perimeter length of the EL membrane was measured. Then, the length of the EL membrane that was closely apposed to an ER membrane (~30 nm, as per the definition of membrane contact sites) was measured in the same sections. Total EL perimeter and EL-ER contact lengths across all sections that captured a single EL were summed up. The sum of the EL-ER contact lengths was divided by the sum of the EL perimeter lengths and multiplied by 100 to give the percentage of total EL surface that contacts the ER. These values were averaged for 5-10 ELs per cell and the data were plotted for 8 cells per condition, conducted across two independent experiments. The results

showed a ~2-fold decrease in the EL-ER contacts in SNX19-KO relative to WT cells, thus demonstrating that SNX19 contributes to the overall extent of EL-ER contacts in the cell.

We also used a proximity ligation assay (PLA)³ to quantify EL-ER contacts in WT and SNX19-KO cells. This was done using antibodies to the endogenous EL protein LAMP1 and endogenous ER protein calnexin, followed by PLA probes. The results showed fewer fluorescent puncta in SNX19-KO relative to WT cells, which are a readout for the close proximity of ELs and ER in these cells (**new Figure 5c,d**).

- *SNX19 colocalisation with LAMP1 is used throughout as a measure of contact sites. To confirm the presence of ER at these sites (as opposed to the all be it quite remote possibility of an endosomal population of SNX19), an additional ER marker should be used. That VAP is excluded from these sites reiterates the need for another ER marker.*

To address this comment, we expressed the ER marker Halo-SEC61B along with SNX19-GFP and LAMP1-RFP and performed live-cell video microscopy. Consecutive frames showed that Halo-SEC61B is indeed present, though not concentrated, at SNX19-GFP–LAMP1-RFP foci, and that SNX19-GFP at these foci is continuous with the faint SNX19-GFP reticular signal corresponding to the ER network (**new Figure 1g**).

Similar observations were made for the ER marker BFP-KDEL (blue fluorescent protein appended with an ER-retrieval signal), which was also present, but not concentrated, at foci containing full-length SNX19-GFP or the hypertethers SNX19^{ΔPXA}-GFP or SNX19¹⁻⁶⁵⁹-GFP together with LAMP1-RFP (**new Figure 3d**).

I also have the following comments:

1. *P4 line 113, not really regulating EL positioning and motility, but maybe interaction of retromer subunit SNX2 with VAP at ER contact sites with endosomal budding sites (Dong et al, 2016) should be mentioned here?*

SNX2 and its interaction with VAP were already mentioned in the Discussion section, but we have expanded the description of its role at EL-ER contact sites in the new version of the paper.

2. *P6, Figure 1: would like to see confirmation that the puncta are contact sites by EM, or at the very least by co-staining with an alternative ER marker.*

The CLEM data already showed that SNX19 hypertethers are present at sites of close apposition of ELs and ER (**Figure 4 and Extended Data Figure 4**). We attempted cryo-immunogold labeling and EM of SNX19, but observed labeling throughout the ER and the contact sites were difficult to make out without the benefit of a fluorescent signal. Instead, and as suggested by the reviewer, we now show by fluorescence video microscopy that the ER markers Halo-SEC61B and BFP-KDEL are present at and continuous with SNX19-GFP–LAMP1-RFP foci (**new Figures 1g and 3d**).

3. *Colocalisation between RAB5 (early endosomes) and LAMP1 (late endosomes and lysosomes) is surprising and the duration of the dextran pulse-chase should be given in the legend and/or text.*

Although most LAMP1 localizes to late endosomes and lysosomes, it also localizes to early endosomes (particularly when overexpressed), as shown by Judith Klumperman's group⁴. It's important to point out that the RAB5A imaged in our experiments was endogenously tagged with Halo, and therefore not overexpressed. The association of LAMP1-RFP with a population of RAB5A-positive early endosomes is thus quite certain.

The duration of the dextran pulse-chase (16-h pulse followed by a 2-h chase), which was indicated in the Methods section, is now also mentioned in the Figure legend of Fig. 1h.

4. Fig 2: Would be nice to shown (absence of) purified mutant (R582) PX interaction with PI3P on PIP strip

The **new Figure 2a,b** shows that mutation of R582 in the PX domain of SNX19 abrogates binding to PI(3)P.

5. Fig 3: Again, need to show an additional ER marker.

In the **new Figure 3d**, we show that the ER marker BFP-KDEL is present, though not enriched, at foci containing GFP-tagged full-length, Δ PXA or 1-659 SNX19 together with LAMP1-RFP. The images also show that the SNX19-GFP foci are continuous with the ER network labeled with BFP-KDEL. These findings support the notion that SNX19 is an ER-resident protein that accumulates at EL-ER contact sites.

6. P8, line 234: I don't think it has been established that SNX19 mediates ER-EL contacts. Fig 4 indicates enrichment at the contact but to say it actively mediates the contact, you need to show that ER-EL contacts are reduced/increased on SNX19 loss/overexpression respectively.

Points 6 and 7 are addressed together below.

7. Fig 5: The effect of SNX19 on endosome distribution is striking but to link it better to contact sites, reduced contacts in the knockdown cells / increased contact in the SNX19-GFP rescue cells needs to be demonstrated.

We thank the reviewer for bringing up this important point. As mentioned above, we now show that SNX19 KO decreases overall EL-ER contacts, as determined by quantitative EM and PLA (**new Figure 5a-d**).

8. Fig 7: It would be nice to see SNX14 staining with ELs and LDs here as well. Also I find it hard to see the blue-green colocalisation.

The localization of SNX14 to LDs after treatment with oleic acid has been extensively demonstrated by Mike Henne's lab⁵ and confirmed by us, but we would prefer not to use precious space to document this finding again.

We have changed the blue in the micrograph to magenta, which we believe is more visible against the green SNX19 signal (**new Figure 7f**).

9. P12, line 367-373: I couldn't see any evidence of lipid droplets at SNX19-positive contacts in the CLEM images, but are the small electron-dense organelles lysosomes (esp evident in extended data fig 3)? I'm not sure why they would be so small - do these structures contain BSA-gold (hard to see from the pdf)? If so, these images may suggest a role for SNX19 in the regulation of endosome-lysosome fusion?

The small electron-dense organelles in the CLEM images are indeed lipid droplets, which are strongly stained by binding of the osmium fixative to unsaturated acyl chains in lipid esters⁶. They do not contain internalized BSA-gold. These LDs are now marked by white arrows and mentioned in the legends of **Figure 4** and **Extended Data Figure 4**.

10. P13, SNX13: Maybe beyond the scope, but it would be nice to show if SNX13 does localise to ER-

endosome contacts – would further strengthen the importance of the PI3P-binding of SNX19 in its role at EL contact sites if it does.

We now show in the **new Extended Data Figure 6f** and mention in the Discussion that SNX13-GFP also co-localizes with LAMP1-RFP, further supporting the importance of PI(3)P binding for contacts with ELs. The expression level of this construct, however, was much lower than that of SNX19-GFP, precluding a detailed characterization of its role in EL-ER contact formation.

Reviewer #2:

In this manuscript Saric and colleagues report that SNX19, one of the mammalian orthologues of Mdm1/SNZ, localizes to endoplasmic reticulum (ER) in contact with endolysosomes (EL). They show that the PX domain of SNX19 binds PI3P and inhibition of PI3P synthesis decrease the co-localization of SNX19 with LAMP1-positive EL, while the increase in PI3P increases this co-localization. They also found that deletion of the SNX19 PXA and PXC domains increase the occurrence of ER-EL contacts, suggesting that these domains play a regulatory role. Some hypotheses are discussed to explain their putative role, however the mechanisms underlying such role are not addressed in this study. The most interesting finding of this study is that SNX19 regulates EL positioning and motility, representing the first example of a SNX with this function. Knock-out of SNX19 results in dispersion of EL and in the increase in mobile (here defined as “free” and “plus directed”) vs “confined” EL. Yet, this function and the possible molecular mechanism involved are not further investigated. The authors conclude showing that, upon oleic acid treatment, SNX19 can also localize in proximity to lipid droplets (LDs), and hypothesize that SNX19 might constrain EL motility to position EL in proximity to LDs to maintain lipid homeostasis. However, the existence of a tripartite ER-EL-LD association mediated by SNX19, despite being potentially very interesting, is not convincingly supported by the preliminary low-resolution microscopy analysis presented here.

We appreciate this reviewer’s comment that “The most interesting finding of this study is that SNX19 regulates EL positioning and motility, representing the first example of a SNX with this function.” That is certainly the main message of our study. We addressed other comments as described below.

A role for SNX19 in the formation of a tripartite ER-EL-LD complex was just mentioned as a possibility, based on the ability of SNX19 to mediate interactions of the ER with both ELs and LDs. The aim of our work was to investigate the role of SNX19-mediated interactions of the

ER with ELs in the regulation of EL positioning and motility, which I think we have convincingly demonstrated. Nevertheless, we did quantify the percentage of SNX19-GFP-positive and SNX19-GFP-negative BODIPY-labeled lipid droplets having associated LAMP1-RFP in the same cells after treatment with oleic acid. We found that, although there was a trend towards more association of SNX19-GFP-positive lipid droplets with LAMP1-RFP, the differences did not rise to the level of statistical significance (see figure on the left). In light of these results, we have removed from the paper speculation about a possible role for SNX19 in ER-EL-LD tripartite complex formation, and just mention that SNX19 may play dual roles in ER-EL and ER-LD interactions.

Major points:

1. The authors show that SNX19 localizes to ER elements closely associated to EL by live cell confocal imaging and also by SIM coupled with CLEM (in the specific case of the PXA deletion mutant). However, to prove with no-doubt that SNX19 is really a tether of ER and EL it should be shown that

SNX19 is indeed localized at sites of apposition between the membranes of such organelles (i.e. by APEX-EM, immuno-EM, etc) and that its overexpression or depletion regulates the extent of ER-EL contacts.

For instance, is there a decrease in the extent of ER-EL contact sites upon SNX19 knockdown? Or an increase upon its overexpression?

We attempted to localize SNX19 to EL-ER contact sites by EM of immunogold-labeled cryo-sections. Unfortunately, these attempts were unsuccessful. Gold labeling was observed throughout the ER (as expected), but higher concentration of gold particles at sites of ER-EL contacts was difficult to make out, probably because of the low sensitivity of the immunogold technique and the fact that SNX19-mediated contacts are not so numerous at steady state. The CLEM data presented in Figure 4 and extended Data Figure 4 are the closest we've come to demonstrating the presence of SNX19 hypertethers at ER-EL contact sites, and we think they are very convincing. We invite the reviewer to look again at these figures and appreciate the fact that the SNX19 foci are found at sites of close apposition of ER with ELs.

We did succeed, however, in assessing the effect of SNX19 KO on the overall extent of ER-EL contacts in the cell. As explained in detail in the response to Reviewer #1 (see above), we used two methods, quantitative transmission EM and a proximity ligation assay (PLA), to demonstrate that SNX19 KO decreases overall ER-EL contacts. The results of these analyses are shown in the **new Figure 5a-d**.

*2. The authors show a novel role (among other SNXs) of SNX19 in restricting endolysosome motility. However, how Snx19 regulates the motility of endosomes remains unclear.
-Is the effect on endolysosome motility directly mediated by ER tethering of EL?*

The fact that the hypertethering SNX19 mutants Δ PXA and 1-659 (**Figure 3**) decrease the motile population of ELs (**Figure 6e,f**) is good evidence that SNX19-mediated tethering regulates EL motility. To bolster this evidence, in the new version of the manuscript we show data demonstrating that SNX19-positive ELs are less motile than SNX19-negative ELs in the same cells and the same field of view (**Figure 6g,h**).

In addition, and as described in the response to Reviewer #1, we considered the alternative possibility that SNX19 regulates EL positioning and motility through interaction with dynein-dynactin or kinesin-1/3 motors, both of which have been previously shown to move LAMP-1-positive organelles. Pulldown experiments showed that GST fused to the dynein light intermediate chain C-terminal domain (which mediates interactions with various dynein adaptors) brought down the known RILP dynein adaptor but not SNX19 (**Extended Data Figure 6e**). In addition, co-immunoprecipitation experiments showed that the kinesin-1 heavy chain KIF5B and light chain KLC2 came down with GFP fused to the known kinesin-1 adaptor SKIP but not to SNX19 (**Extended Data Figure 6c**). We also could not detect interaction of SNX19 with the kinesin-3 KIF1B, another motor of LAMP1-positive organelles, although we lacked a positive control for this particular experiment (**Extended Data Figure 6d**). These results make it unlikely that SNX19 regulates EL positioning and motility through interactions with microtubule motors.

The authors show that the ER morphology and distribution in SNX19 KO cells is not affected. How this reconciliates with the EL motility defect? Are the "free" or "directed" endosomes in Snx19 KO cells still in contact with ER?

One possible and easiest way to address this point could be to co-express fluorescently tagged LAMP1 and an ER marker to follow their dynamics and association in SNX19 KO cells.

As the reviewer correctly points out, we did not observe gross alterations in ER morphology and distribution in SNX19-KO relative to WT cells. In addition, we did not find obvious differences in ER dynamics in WT and SNX19-KO cells co-expressing GFP-SEC61B (to label ER) and LAMP1-RFP (to label ELs). It should be noted, however, that these observations were made in the peripheral ER and not in the perinuclear ER, where most SNX19-mediated ER-EL interactions take place but is more difficult to image.

We did examine the extent of overlap of static, SNX19-GFP-positive ELs and surrounding motile, SNX19-GFP-negative ELs to the ER in the peripheral cytoplasm. These experiments showed that whereas static, SNX19-positive ELs contacted the ER 100% of the time, motile SNX19-negative ELs were associated with the ER ~80% of the time (**Extended Data Figure 6a,b**). Thus, SNX19 recruitment to ELs increases their apparent association with the ER. However, it should be borne in mind that peripheral ER tubules are closely aligned with microtubules, so, at the resolution of fluorescence microscopy, it's not possible to distinguish EL association with the ER from association of both ELs and ER with the same microtubule tracks. These data and associated caveats are now mentioned in the text.

-Is SNX19 interacting with any rab or kinesin regulating endosomal movement?

We have not screened for Rabs that interact with SNX19; this remains a goal for future studies. As discussed above, we did test for interaction of SNX19 with kinesin-1 and kinesin-3 (both involved in the movement of LAMP1-positive organelles) and with dynein. The results of these experiments were negative. These findings are now shown in the **new Extended Data Figure 6c-e** and discussed in the text.

-Does SNX19 regulate EL lipid composition and consequently recruitment of motor or other effector proteins?

We have not examined the possible role of SNX19 in regulating the lipid composition of ELs, because we think that this exceeds the scope of our study. Furthermore, as mentioned above, we have no evidence that SNX19 regulates the recruitment of motor molecules to ELs.

3. The last figure (Fig 7) of the manuscript compares SNX19 and SNX14 localization, showing that SNX14 does not form contacts with endolysosomes. I find this comparison very pertinent, however I would move it ahead in the manuscript (linked to Figure1 or in Supplementary material).

We thank the reviewer for this suggestion, but we would prefer to keep this finding at the end of our study because we feel that the story flows better this way.

In this figure the authors also show that SNX19 can additionally localize to ER elements in proximity to LDs, as indicated by the partial overlap of SNX19 and GFP-ADRP (a LD surface protein)(Fig7d). However, the image shown is not very convincing, and the shape of LDs not easy to see. I would suggest using in parallel LD markers such as BODIPY or LTox

We now include images of SNX19-GFP plus and minus oleic acid, in which LDs are labeled with monodansylpentane (MDH), a commonly used LD neutral lipid stain⁷ (**Extended Data Figure 7**).

Electron microscopy analysis should also be performed to confirm that these structures are indeed ER-LD contact sites and to quantify their extent vs the ER-EL contacts.

As per the editor's suggestion, we have not performed these analyses, since it exceeds the scope of our study, which is to show a role of SNX19 in the regulation of EL positioning and motility.

In the same figure (Fig 7f) the authors also try to prove the existence of a tripartite contact site association ER-EL-LD mediated by SNX19, but is too difficult to see this association in the images shown, probably given the resolution limits of the approach used and also because the inset images are too small, the individual channel of the entire cell are not shown and quantifications of the occurrence and extent of this tree-way organelle association are lacking (i.e. how many of the ER-EL contacts are also associated to LDs?). To prove the existence of such tripartite association a deeper morphological characterization would be needed, and their functional relevance should be addressed, but I am not sure that this is within the scope of this manuscript.

We agree with the reviewer that demonstrating the existence of tripartite ER-EL-LD interactions is beyond the scope of our study. We only mentioned it as a possibility. We have now removed this speculation from our paper and only mention that SNX19 may play dual roles in ER-EL and ER-LD contacts.

Minor points:

- Some images seem to be overexposed (e.g., in Figures 1b, 1c, 3b, 7d)

We have adjusted the intensity/contrast of micrographs in Figures 1b, 1c, 3b and 7d.

Extended Data Figure 7 additionally has unsaturated micrographs of MDH as a LD marker.

- How the authors explain why the TM deletion mutant of SNX19 (Figure 1c) is not binding, at least in part, the endolysosomes (as the PX domain of SNX19 has been shown to bind PI3P)? If PXA domain inhibits PI3P binding, maybe a doubly-deleted mutant (SNX19 Δ PXA Δ TM) would bind the endolysosomes?

In response to this comment, we generated a new construct expressing the doubly-deleted SNX19 mutant suggested by the reviewer (Δ TM Δ PXA). We found that this mutant indeed localizes to ELs (**Extended Data Figure 3d**), further supporting the finding that the PXA domain inhibits SNX19 binding to EL.

- In Figure 4b1 Snx19 Δ PXA fluorescence seems to be also inside endosomes. Is this area of interest an individual focal plane?

Since this is a CLEM image, the reviewer is correct that there is a small amount of SNX19^{APXA}-GFP signal inside the EL in **Figure 4b**. This is not a frequent finding and may reflect a small level of microautophagy taking place in these cells. Notice that other images in Figure 4 and Extended Data Figure 4 do not show intraluminal SNX19^{APXA}-GFP.

-Method section: How do the authors analyze the colocalization of SNX19 and LAMP1 in fixed cells? How is the cell surface visualized in the EL peripheral dispersal measurements?

We appreciate both of these questions. We now explain in the Methods section how both of these analyses were conducted:

“For experiments comparing SNX19-positive and SNX14-positive EL contacts (Fig. 7a-c), fixed cells were imaged as z-stacks. Cells were analyzed by zooming in and thresholding the image as described above to identify higher-intensity SNX19 or SNX14 puncta (higher than

the local ER-resident reticular signal) that colocalized with LAMP1-RFP structures. A true contact could usually be observed in two consecutive z-slices.”

“Cell outlines were traced in Fiji/ImageJ by briefly boosting the LAMP1 signal such that cytosolic fluorescence could be visualized. The LAMP1 signal was returned to original settings, a threshold was applied to eliminate background and the total cellular LAMP1 signal was measured.”

-Some typo: Page 7, line 202: “SNX19” instead of “SNAX19”; s23 line 737: “significance” instead of “ignificance”

Corrected.

Reviewer #3:

The authors characterize the unconventional sorting Nexin SNX19, which harbors two transmembrane domains in addition to the hallmark PX domain. The authors show that overexpressed SNX19 localizes to ER-EndoLysosome (ER-EL) contact sites, where it restricts EL mobility, probably through tethering the ELs to the less mobile ER. Overall, the study is well written and the data are of high technical quality. The assays are convincing and I largely agree with the authors’ conclusions.

I have only one major concern regarding the interpretation of data. Throughout the study, the authors simply assume that SNX19 mediates ER-EL tethering. While I also think that this is the case, this is not really proven by any experiment. SNX19 could just localize to ER-EL contacts which are established by other proteins such as Protrudin and RAB7.

It would be great if the authors could show that ER-EL contact sites are altered in their SNX19 KO cells. In theory, there should be less or weakened (more transient?) contact sites. Conversely, overexpression of SNX19 should induce ER-EL contact sites. This could be addressed by FRET analysis between Calnexin and LAMP1, just as an example. Any other assay that can visualize or quantify ER-EL contact sites could also be used instead of FRET analysis.

Overall, I am supportive of publication in Nature Communications as I think that this is an important and interesting study that will garner a lot of interest from the cell biological community. However, the authors should really show that SNX19 does indeed mediate and/or regulate ER-EL contacts.

We thank this reviewer for concluding that “the study is well written and the data are of high technical quality. The assays are convincing and I largely agree with the authors’ conclusions.” and that “this is an important and interesting study that will garner a lot of interest from the cell biological community.”

The reviewer highlights a key point that we wholeheartedly agree needed to be addressed: whether SNX19 influences the overall number of EL-ER contacts. As described in the response to Reviewer #1, we now show that SNX19 KO decreases the overall extent of EL-ER contacts, as determined by quantitative EM and PLA (**new Figure 5a-d**).

In addition, I have listed some more minor points below.

Additional points:

-Could the authors co-stain SNX19-GFP with some endogenous markers such as EEA1, LAMP1, RAB7? This would corroborate the localization data.

We now show that SNX19-GFP is indeed found at contacts with the endogenous late endosomal/lysosomal marker LAMP1 and early endosomal marker EEA1 (**Extended Data Figure 1a**), as well as the early endosomal marker Rab5 (detected in Rab5-Halo knock-in cells) (**Extended Data Figure 1c**). We attempted staining for endogenous Rab7, but there was too much background for accurate assessment of co-localization.

-All the data on SNX19 localization to ELs is derived from overexpressed protein. In the absence of an antibody, can the authors at least overexpress SNX19-GFP (or other tag) at the lowest possible levels in their SNX19 KO cell line and demonstrate that the SNX19-LAMP1 colocalization can still be observed?

To address this comment, we co-transfected SNX19-GFP and LAMP1-RFP into SNX19-KO cells and then imaged the lowest expressing cells. Despite the low SNX19-GFP signal, we could still observe foci containing SNX19-GFP and LAMP1-RFP (see figure below) (for the reviewer only).

-Figure 4: Why did the authors use a mitochondria marker in this figure? To show that SNX19 does not establish mitochondria contacts? The authors should explain why they used a mitochondria marker

We now clarify the purposes of labeling mitochondria in the CLEM experiments. In the main text within the Results section we state “Labeling mitochondria with Mito-BFP served two important purposes: 1) it facilitated manual alignment of fluorescence and EM images, as these organelles are large and easily identifiable landmarks, and 2) they were used as an internal specificity control for SNX19 contacts, which occurred with LAMP1-RFP but not Mito-BFP labeled organelles.”

References

- 1 Jordens, I. *et al.* The Rab7 effector protein RILP controls lysosomal transport by inducing the recruitment of dynein-dynactin motors. *Curr Biol* **11**, 1680-1685 (2001).
- 2 Schroeder, C. M., Ostrem, J. M., Hertz, N. T. & Vale, R. D. A Ras-like domain in the light intermediate chain bridges the dynein motor to a cargo-binding region. *Elife* **3**, e03351, doi:10.7554/eLife.03351 (2014).
- 3 Soderberg, O. *et al.* Direct observation of individual endogenous protein complexes in situ by proximity ligation. *Nat Methods* **3**, 995-1000, doi:10.1038/nmeth947 (2006).
- 4 Fermie, J. *et al.* Single organelle dynamics linked to 3D structure by correlative live-cell imaging and 3D electron microscopy. *Traffic* **19**, 354-369, doi:10.1111/tra.12557 (2018).

- 5 Datta, S., Liu, Y., Hariri, H., Bowerman, J. & Henne, W. M. Cerebellar ataxia disease-associated Snx14 promotes lipid droplet growth at ER-droplet contacts. *J Cell Biol*, doi:10.1083/jcb.201808133 (2019).
- 6 Cheng, J. et al. Quantitative electron microscopy shows uniform incorporation of triglycerides into existing lipid droplets. *Histochem Cell Biol* **132**, 281-291, doi:10.1007/s00418-009-0615-z (2009).
- 7 Chen, B.-H., Yang, H.-J., Chou, H.-Y., Chen, G.-C. & Yang, W. Y. in *Histochemistry of Single Molecules: Methods and Protocols* (eds Carlo Pellicciari & Marco Biggiogera) 231-236 (Springer New York, 2017).

REVIEWERS' COMMENTS

Reviewer #1 (Remarks to the Author):

The manuscript looks great - all my concerns have been thoroughly addressed. The new data in Figure 5 is especially nice and robustly demonstrates a role for SNX19 in ER-EL contact site formation.

However, while it is not at all important to the focus of the paper, I disagree with the authors that the organelle labelled with a white arrowhead in Figure 5b (and supplemental fig. 5) is a LD. To me this looks like part of an EL (zooming into the high z image in 5b, small ILVs can be discerned) that would likely be connected with the BSA-Au containing EL in a different focal plane, ie, it looks separate because of the angle of the sectioning. In Figure 5b, I think it even overlays with the LAMP staining. LDs tend to be bigger than this and unless the cells have been treated with FAs, the electron density of the LD core tends to be relatively low and uniform. This is just my opinion - it's your call!

Reviewer #2 (Remarks to the Author):

The major finding of this study is the novel role of SNX19 in regulating EL positioning and motility, that represents an important advance for the cell biology field.

In this revised version of the manuscript the authors provide additional evidence that SNX19 regulates the extent of ER-EL contacts, supporting their hypothesis that loss of ER-EL contacts underlies the peripheral redistribution of endosomes in SNX19 KO cells.

The authors removed from the manuscript the speculation of a possible role of SNX19 in tripartite ER-EL-LD for the lack of sufficient evidence supporting this model.

Overall, I consider that they sufficiently addressed or discussed my major points, although the mechanisms underlying SNX19 functions in regulating EL positioning and motility remain still to be dissected, possibly in another study as this seems behind the scope of this paper. By this is not a criticism, it just reflects my curiosity to hear more about SNX19 role at ER-EL contact sites, as I consider it very interesting. Thus, I support the publication of this manuscript on Nature Communication.

Reviewer #3 (Remarks to the Author):

The authors have performed two independent assays that confirm an active role of SNX19 in the formation of EL-ER contact sites. Therefore, the authors have addressed my main concern and I have no further reservations regarding publication in Nature Communications.

Responses to Reviewers

Reviewer #1 (Remarks to the Author):

The manuscript looks great - all my concerns have been thoroughly addressed. The new data in Figure 5 is especially nice and robustly demonstrates a role for SNX19 in ER-EL contact site formation.

However, while it is not at all important to the focus of the paper, I disagree with the authors that the organelle labelled with a white arrowhead in Figure 5b (and supplemental fig. 5) is a LD. To me this looks like part of an EL (zooming into the high z image in 5b, small ILVs can be discerned) that would likely be connected with the BSA-Au containing EL in a different focal plane, ie, it looks separate because of the angle of the sectioning. In Figure 5b, I think it even overlays with the LAMP staining. LDs tend to be bigger than this and unless the cells have been treated with FAs, the electron density of the LD core tends to be relatively low and uniform. This is just my opinion - it's your call!

We thank the reviewer for complimenting our additional data that demonstrates a role for SNX19 in ER-EL contact site formation.

With regards to the reviewer's comment about the identity of the structure in Fig. 4b and Supplementary Fig. 4a (white arrowheads), we appreciate the comments made; however, upon consultation with our EM expert and upon examination of transmitted light images of these cells (not included in the manuscript), we still believe these structures to be LDs. There are several reasons for this: 1) corresponding transmitted light images show an intensely black/white contrast dot in the same spot as these structures, which is typically what LDs look like under transmitted light when adjusting focus, 2) After looking through the corresponding EM sections, we could not see any gold inside these structures, whereas BSA-gold typically aggregates within lysosomes, and 3) We believe the ILVs the reviewer mentions in the high z image in Fig. 4b appear that way because the image is a bit "grainy". Due to their extremely thin nature, these sections were of somewhat low contrast, thus contrast adjustments to better visualize the membranes resulted in an overall grainy appearance. We very much thank the reviewer for their comments, but we have opted, at the suggestion of our EM expert, to leave this part as is.

Reviewer #2 (Remarks to the Author):

The major finding of this study is the novel role of SNX19 in regulating EL positioning and

motility, that represents an important advance for the cell biology field.

In this revised version of the manuscript the authors provide additional evidence that SNX19 regulates the extent of ER-EL contacts, supporting their hypothesis that loss of ER-EL contacts underlies the peripheral redistribution of endosomes in SNX19 KO cells.

The authors removed from the manuscript the speculation of a possible role of SNX19 in tripartite ER-EL-LD for the lack of sufficient evidence supporting this model.

Overall, I consider that they sufficiently addressed or discussed my major points, although the mechanisms underlying SNX19 functions in regulating EL positioning and motility remain still to be dissected, possibly in another study as this seems behind the scope of this paper. By this is not a criticism, it just reflects my curiosity to hear more about SNX19 role at ER-EL contact sites, as I consider it very interesting. Thus, I support the publication of this manuscript on Nature Communication.

We are pleased with the reviewer's assessment of our revised manuscript and that they find the study interesting. We hope to uncover more details of the mechanisms behind SNX19 tethering function in future studies.

Reviewer #3 (Remarks to the Author):

The authors have performed two independent assays that confirm an active role of SNX19 in the formation of EL-ER contact sites. Therefore, the authors have addressed my main concern and I have no further reservations regarding publication in Nature Communications.

We thank the reviewer for their comments and we are pleased that they consider our revised manuscript suitable for publication in Nature Communications.